# ELEMENTAL: INTERACTIVE LEARNING FROM DEMONSTRATIONS AND VISION-LANGUAGE MODELS FOR REWARD DESIGN IN ROBOTICS

## ABSTRACT

Reinforcement learning (RL) has demonstrated compelling performance in robotic tasks, but its success often hinges on the design of complex, ad hoc reward functions. Researchers have explored how Large Language Models (LLMs) could enable non-expert users to specify reward functions more easily. However, LLMs struggle to balance the importance of different features, generalize poorly to out-of-distribution robotic tasks, and cannot represent the problem properly with only text-based descriptions. To address these challenges, we propose ELEMENTAL (intEractive LEarning froM dEmoNstraTion And Language), a novel framework that combines natural language guidance with visual user demonstrations to align robot behavior with user intentions better. By incorporating visual inputs, ELEMENTAL overcomes the limitations of text-only task specifications, while leveraging inverse reinforcement learning (IRL) to balance feature weights and match the demonstrated behaviors optimally. ELEMENTAL also introduces an iterative feedback-loop through self-reflection to improve feature, reward, and policy learning. Our experiment results demonstrate that ELEMENTAL outperforms prior work by 42.3% on task success, and achieves 41.3% better generalization in out-of-distribution tasks, highlighting its robustness in LfD.

## 1 INTRODUCTION

Reinforcement Learning (RL) has been shown to be a powerful tool for enabling robots to perform complex tasks across a wide range of domains, from manipulation (Kober et al., 2013; Levine et al., 2016) to navigation (Tai et al., 2017; Zhu et al., 2017). However, the effectiveness of RL hinges on the availability of a carefully designed reward function that accurately encapsulates the desired behavior. Without sophisticated reward functions, RL agents often struggle to learn competent policies (Matignon et al., 2006); worse yet, poorly designed reward functions can lead RL agents to achieve undesirable outcomes (Gupta et al., 2024). Booth et al. (2023) shows reward specification is non-trivial even for experts, and designing reward functions that align with end-users' expectations is particularly challenging due to the varied and latent user preferences (Abouelazm et al., 2024).

Considering recent advancements in large models (e.g., large language models (LLMs) and vision-language models (VLMs)) on text understanding (Bommasani et al., 2021; Touvron et al., 2023) and emergent abilities (Kojima et al., 2022; Wei et al., 2022), researchers have explored utilizing LLMs for reward engineering. For instance, the EUREKA framework takes as input a text description of the task, queries LMs for a draft of the reward function, and trains a policy with the reward function (Ma et al., 2023). This paradigm, while promising, presents several limitations. First, describing complex robotic tasks purely through language is imprecise: humans often have latent, unspoken preferences that are difficult to articulate fully, leading to incomplete or ambiguous descriptions of their objectives (Nisbett & Wilson, 1977; Ericsson & Simon, 1980; Hoffman et al., 1995; Feldon, 2007). Second, even if all objective function components are accurately conveyed, determining the relative importance or weights of these components poses another significant challenge: assigning these weights involves subtle mathematical trade-offs, something that LLMs are not particularly equipped to handle. As a result of these limitations, methods like EUREKA struggle to generalize well to out-of-distribution tasks.

Given these limitations, a more natural and effective approach is for users to provide demonstrations of the desired behavior to supplement a general task description. Demonstrations offer rich, illustrative information that can capture not only the task objectives but also the nuanced, latent preferences that may be difficult to express verbally. Learning from Demonstration (LfD) approaches seek to leverage human-provided demonstrations to reverse-engineer the underlying objective and optimize a policy accordingly (Chen et al., 2020; 2022; Suay et al., 2016; Ravichandar et al., 2020). However, a key challenge in LfD is the ambiguity in interpreting demonstrations – there can be an infinite number of possible reward functions that could explain the same set of demonstrations, a problem commonly referred to as the *reward ambiguity problem* in Inverse Reinforcement Learning (IRL) (Abbeel & Ng, 2004). Prior methods have sought to address this ambiguity by pre-designing features to constrain the space of possible reward functions, but this often limits the flexibility and generalizability of the learned rewards and policies (Zhu & Hu, 2018; Arora & Doshi, 2021). Our key insight is that language models are well-suited to contextualize demonstrations and infer task features, narrowing down the possible interpretations and enabling robots to learn more robustly.

We propose to integrate the strengths of language models and LfD methods, leveraging (1) the emergent reasoning capabilities of language models to identify robust and relevant objective function components, and (2) the demonstration-matching capabilities of LfD to determine the optimal weighing of these components. Crucially, we incorporate visual demonstrations into Vision-Language Models (VLMs), facilitating a more comprehensive understanding of human objectives. Additionally, we introduce a self-reflection mechanism that enables VLM and LfD to iteratively improve both the feature extraction and the reward & policy learning. This fusion of LfD and VLMs offers a novel pathway for more effective robotic learning from demonstrations. This paradigm also mirrors how humans naturally learn from others. When observing a demonstration, humans typically (1) identify the key aspects of the task, (2) formulate a policy to match the demonstration based on those salient features, and (3) reflect on the discrepancies between their own behavior and the demonstration to refine their understanding and execution (Locke, 1987; Di Stefano et al., 2014). By iterating through these steps, humans progressively improve their performance. This iterative cycle of observation, reflection, and refinement is not only fundamental to human learning but also serves as an ideal framework for robotic learning (Chernova & Thomaz, 2014).

To develop this novel integration between VLM and LfD, we introduce ELEMENTAL (intEractive LEarning froM dEmoNstraTion And Language). ELEMENTAL enables robots to identify key task features from human demonstrations, learn rewards and policies that align on these features, and iteratively reflect on their performance to improve over time. Our key contributions are three-folds:

1. We propose a novel, general framework that integrates VLMs and LfD, introducing an iterative self-reflecting mechanism for continuous improvement. ELEMENTAL is the first to incorporate visual demonstration inputs into language models to accomplish LfD, which facilitates more accurate behavior understanding within demonstrations.

2. We evaluate ELEMENTAL on a set of challenging, standard robotic benchmarks in IsaacGym, demonstrating its superior performance over previous state-of-the-art (SOTA) reward design and LfD methods by 42.3%, showcasing its effectiveness.

3. We further assess ELEMENTAL's generalization capabilities by designing novel variants of the standard benchmarks. Our results show that ELEMENTAL achieves 41.3% better generalization than existing methods, underscoring the importance of combining VLMs with LfD.

## 2 RELATED WORK

**Learning from Demonstration (LfD) –** LfD approaches, such as Behavior Cloning (BC) and IRL, have long been used to enable robots to learn from human-provided demonstrations. BC (Ross et al., 2011), a supervised learning approach, is effective for relatively simple tasks, but it is prone to compounding errors (known as *covariate shift*). IRL (Ng & Russell, 2000; Abbeel & Ng, 2004; Ziebart et al., 2008; Ziebart, 2010) seeks to infer the underlying reward function that explains the demonstrated behavior. However, reward ambiguity poses a significant challenge – an infinite number of reward functions could explain the same behavior. This challenge becomes exacerbated in complex domains with limited, heterogeneous demonstrations Chen et al. (2020); Peng et al. (2024a). ELEMENTAL addresses this key limitation by integrating VLMs to inject emergent reasoning capabilities into the learning process. VLMs reduce ambiguity by providing semantic context that allows robots to better understand relevant task features.

**Language Models as Reward Engineers –** Recent works, such as EUREKA (Ma et al., 2023) and L2R (Yu et al., 2023), leverage LLMs to convert language descriptions into reward functions, offering a promising alternative to manual reward engineering. However, these methods are limited by their reliance solely on brief task descriptions, restricting their ability to capture the full complexity of robotic tasks and the subtle preferences of users. Additionally, determining the appropriate weighting of different objective function components is particularly challenging, as the reward design process is disconnected from policy training, resulting in poor out-of-distribution generalization. ELEMENTAL addresses these limitations by integrating IRL with VLMs and supplementing task descriptions with demonstrations. In ELEMENTAL, the responsibility of assigning reward component weights is shifted from the VLM to IRL, which matches the reward components to the demonstrated behaviors. This allows the VLM to focus on its strength—semantic understanding and task feature identification.

A closely related line of research is that by Peng et al. (2024b), which also integrates feature design using LLMs with IRL. While this work demonstrates promising results, it operates under several limiting assumptions. First, it assumes that demonstrations can be provided as input exclusively as text-based state-vector sequences. Due to this constraint, the work is only applied to simpler tasks (e.g., problem horizons of fewer than five steps) and where only one or two features are missing (with other ground-truth features given a priori). Additionally, as seen in other works (Yu et al., 2023; Kwon et al., 2023), this method relies on specially designed prompts tailored to each experimental domain. These limitations hinder its scalability to more complex, high-dimensional tasks typically encountered in real-world robotic applications. In contrast, ELEMENTAL leverages visual modality for demonstration inputs, making it well-suited for more complex tasks where textual descriptions alone are insufficient. In addition, to account for the complex domains, we introduce an upgraded MaxEnt-IRL approach, detailed in Section 4.2. By incorporating visual modality, using a general prompt across domains, and developing enhanced IRL, ELEMENTAL provides a more scalable solution, as demonstrated in our results on standard robotic benchmarks without requiring any prior knowledge of task features.

**LM-Assisted Robot Learning –** Several recent works have sought to leverage LMs to assist in robotic learning, such as RL-VLM-F (Wang et al., 2024), RoboCLIP (Sontakke et al., 2024), and Du et al. (2023). While promising, these methods depend on a surrogate reward derived from the LM's understanding of the task-state alignment, which can introduce inaccuracies. Furthermore, these approaches lack interactivity with users, a key component shown to be helpful in gaining human trust (Chi & Malle, 2024). In contrast, ELEMENTAL potentially allows engineers and users to interactively refine the robot's behavior, ensuring that the robot is user-aligned.

Other works, such as Wang et al. (2023) and Mahadevan et al. (2024), directly query LMs to output robot actions or primitives. These approaches rely heavily on the LLM's ability to plan and optimize actions, but LLMs are not inherently designed for mathematical optimization required for robot control. ELEMENTAL addresses these limitations by using VLMs to understand task features and by deferring the demonstration-to-policy alignment to IRL algorithms, which are better suited to optimize behavior in complex environments.

## 3 PRELIMINARY

In this section, we introduce preliminaries on Markov Decision Process (MDP), Inverse Reinforcement Learning (IRL), and Maximum-Entropy IRL (MaxEnt-IRL).

***Markov Decision Process***: We formulate the robot learning problem as a Markov Decision Process (MDP), defined by the tuple $(\mathcal{S}, \mathcal{A}, T, R, \gamma)$. $\mathcal{S}$ is the set of states the agent can occupy. $\mathcal{A}$ is the set of actions the agent can take. $T : \mathcal{S} \times \mathcal{A} \times \mathcal{S} \to [0, 1]$ represents the transition probability function, where $T(s, a, s')$ gives the probability of transitioning from state $s$ to state $s'$ after taking action $a$. $R : \mathcal{S} \to \mathbb{R}$ is the reward function that assigns a scalar reward to each state. $\gamma \in [0, 1)$ is the temporal discount factor. The goal of Reinforcement Learning (RL) is to learn a policy $\pi : \mathcal{S} \to \mathcal{A}$ that maximizes the expected cumulative reward, given by: $J(\pi) = \mathbb{E}_{\tau \sim \pi} \left[ \sum_{t=0}^{\infty} \gamma^t R(s_t) \right]$.

***Inverse Reinforcement Learning***: In IRL, instead of explicitly programming a robot's behavior, we aim to learn the underlying objective or reward function, $R(s)$, which explains the behavior demonstrated by a human expert. A set of demonstrations are given, $\mathcal{D} = \{\tau_i\}_{i=1}^N$, where each trajectory $\tau_i = (s_1^i, a_1^i, s_2^i, a_2^i, \dots)$ consists of a sequence of states and actions. We assume that

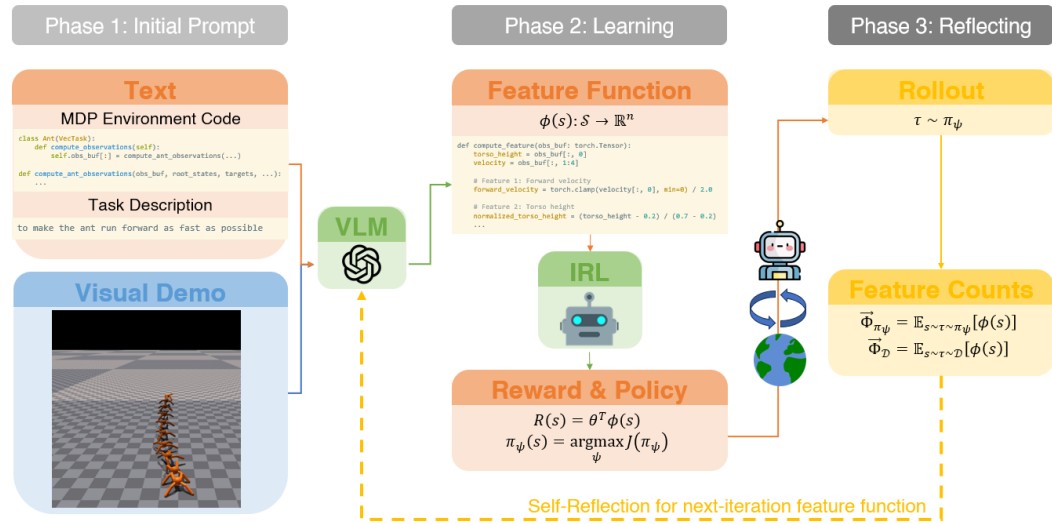

Figure 1: This figure illustrates the overall pipeline of ELEMENTAL. The process begins with an initial prompt to the VLM, which generates a draft of the feature function based on both textual descriptions and visual demonstrations. In the learning phase, ELEMENTAL infers the reward and policy from the drafted feature function and the demonstration. In the final phase, ELEMENTAL performs self-reflection by comparing the feature counts from the generated trajectory and the demonstration, again utilizing the drafted feature function. This self-reflection loop updates the feature function by feeding the results back to the VLM for iterative refinement.

the true reward function, $R(s)$, is a linear combination of features, as shown in Equation 1, where $\phi(s) \in \mathbb{R}^d$ is a feature vector representing the task-relevant or user-preference-relevant properties of the state, and $\theta \in \mathbb{R}^d$ is a weight vector that specifies the relative importance of each feature. The goal of IRL is to recover $\theta$ based on the provided demonstrations $\mathcal{D}$.

$$R(s) = \theta^T \phi(s) \tag{1}$$

MaxEnt-IRL (Ziebart et al., 2008) models the likelihood of a trajectory under the assumption that the expert's behavior is stochastically optimal, as shown in Equation 2. In this equation, $Z(\theta)$ is the partition function, $Z(\theta) = \sum_\tau \exp\left(\sum_{s_t \in \tau} R(s_t)\right)$.

$$P(\tau|\theta) = \frac{1}{Z(\theta)} \exp\left(\sum_{s_t \in \tau} R(s_t)\right) = \frac{1}{Z(\theta)} \exp\left(\sum_{s_t \in \tau} \theta^T \phi(s)\right) \tag{2}$$

The objective of MaxEnt IRL is to find the reward weights $\theta$ that maximize the likelihood of the expert demonstrations, as shown in Equation 3.

$$\hat{\theta} = \arg\max_\theta \sum_{\tau \in \mathcal{D}} \log P(\tau|\theta) \tag{3}$$

In ELEMENTAL, we upgrade MaxEnt-IRL to be suitable for high-dimensional robotic tasks, allowing us to link features from VLMs to demonstrations via the weight vector $\theta$.

## 4 METHOD

In this section, we describe our algorithm, ELEMENTAL, which consists of three interconnected phases, as shown in Figure 1. The first phase involves constructing an initial feature function through a VLM based on visual human demonstrations and environment specifications, detailed in Section 4.1. In the second phase, this feature function is integrated with IRL to learn a reward function and policy that best align with the demonstrated behaviors (Section 4.2). Finally, the third phase introduces a self-reflection mechanism that automatically compares the learned behavior with the demonstrations, enabling iterative refinement of the feature function, the reward function, and the policy (Section 4.3).

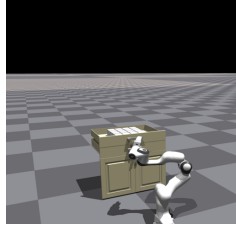

(b) Frame one: start of the trajectory.

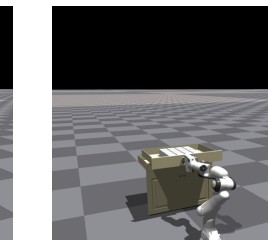

(c) Frame two: reaching the cabinet handle.

(a) Superimposing ant demonstration.

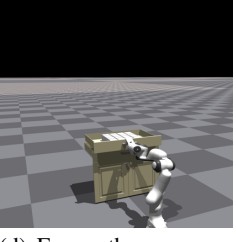

(d) Frame three: opening the cabinet.

(e) Frame four: the cabinet is opened.

Figure 2: This figure illustrates the visual demonstrations for both locomotion and manipulation tasks. (a) shows an example from the Ant locomotion task, where superimposed images are used. For manipulation tasks, superimposed images can result in cluttered robot poses, so we use key frames as visual demonstration inputs instead. (b-e) present four key frames from the FrankaCabinet manipulation task.

### 4.1 PHASE 1: INITIAL PROMPT

The first phase begins with an initial prompt that includes three inputs: (1) the MDP environment code, which specifies the environment's state space $\mathcal{S}$, (2) a text description of the task, and (3) a visual human demonstration of the desired task. The form of the visual demonstration depends on the task domain (Figure 2): for tasks where superimposition is meaningful (e.g., locomotion tasks such as the moving-forward ant or humanoid), we compose a superimposed image that displays the sequence of motions from demonstration (darker shades indicate temporal progression). For manipulation tasks, where superimposition can result in cluttered robot poses (e.g., the FrankaPanda Cabinet), four key frames are provided to illustrate stages of the task execution. By incorporating visual demonstrations alongside language-based task descriptions, ELEMENTAL allows the VLM to generate feature functions that more accurately reflect the user's latent goals.

These inputs are then processed by the VLM, which leverages its emerging reasoning capabilities to infer the task-relevant features. The goal is not simply to describe the state but to capture important factors that align with the user's intent for the task. The output of this phase is a feature function $\phi : \mathcal{S} \to \mathbb{R}^n$, where $\phi(s) = (\phi_1(s), \phi_2(s), \ldots, \phi_n(s))$ represents a vector of features describing the key aspects of the task and $n$ is the number of features. Given VLM's coding capabilities (Piccolo et al., 2023), we ask the VLM to output feature functions as Python codes, providing a structured representation. If the returned code is not executable (e.g., wrong function signature), we re-prompt the VLM with the trackback information up to three times. A successfully executable feature function serves as the foundation for the reward learning and policy optimization in the subsequent phase.

### 4.2 PHASE 2: LEARNING

Once the feature function has been drafted in Phase 1, the next step is to optimize the reward function $R_\theta(s) = \theta^T \phi(s)$ to match the demonstrations, using the feature function $\phi(s)$. As introduced in Section 3, we modify the MaxEnt-IRL algorithm to accomplish reward and policy learning in more complex tasks. We name it Approximate MaxEnt-IRL, as shown in Algorithm 1.

Directly computing the partition function $Z(\theta)$ is intractable due to the summation over all possible trajectories. We circumvent the need for explicit computation of $Z(\theta)$ with the key insight (Ziebart et al., 2008) that the gradient of the MaxEnt IRL objective is given by the feature expectation dif-

---

**Algorithm 1:** Approximate MaxEnt-IRL

---

**Input** : feature function $\phi : \mathcal{S} \to \mathbb{R}^n$, demonstration $\mathcal{D}$, number of IRL iterations $m$, learning rate $\alpha$, policy learning steps $k$

1 Initilize reward function feature weights $\theta = \{1/n\}_{i=1}^n$ and policy weights $\psi$.

2 **for** $i \leftarrow 1$ **to** $m$ **do**

3      **for** $j \leftarrow 1$ **to** $k$ **do**

4          Optimize $\pi_\psi$ based on $J(\pi_\psi)$ with $R_\theta(s) = \theta^T \phi(s)$ via PPO

5      Obtain $\nabla_\theta$ by Equation 5 and normalize to get $\nabla'_\theta$ by Equation 6

6      Update $\theta$ by $\theta \leftarrow \theta + \alpha \nabla'_\theta$

7      Normalize $\theta$ by Equation 7

**Output:** $R_\theta, \pi_\psi$

---

ference between the demonstrated trajectories and the stochastically optimal trajectory under $\theta$, as shown in Equation 4.

$$\nabla_\theta = \mathbb{E}_{\tau \sim \mathcal{D}} \left[ \sum_{s \in \tau} \phi(s) \right] - \mathbb{E}_{\tau \sim P(\tau|\theta)} \left[ \sum_{s \in \tau} \phi(s) \right] \tag{4}$$

We approximate $P(\tau|\theta)$ by a parameterized policy, $\pi_\psi$, that optimizes the estimated reward, $R_\theta(s) = \theta^T \phi(s)$, shown in Equation 5.

$$\nabla_\theta \approx \mathbb{E}_{\tau \sim \mathcal{D}} \left[ \sum_{s \in \tau} \phi(s) \right] - \mathbb{E}_{\tau \sim \pi_\psi} \left[ \sum_{s \in \tau} \phi(s) \right] \tag{5}$$

In particular, we optimize the policy $\pi_\psi$ interleaved with $R_\theta$, following a paradigm similar to that of AIRL (Fu et al., 2018), as shown in Algorithm 1 lines 3-4 and line 6. Intuitively, by applying this gradient, we refine $\theta$ to ensure that the reward function $R_\theta(s)$ more accurately reflects the features emphasized in the demonstration and guide $\pi_\psi$ to align the policy's behavior closer to the demonstrated behavior. The learned policy, $\pi_\psi$, is also a natural by-product of this process, serving as the ultimate goal of LfD and laying the foundation for the reflection phase in Phase 3.

As the feature's numerical magnitude could vary based upon empirical findings, we apply a normalization procedure to the reward gradient to ensure stable learning. First, we normalize the 1-norm of the gradient vector, shown in Equation 6.

$$\nabla'_\theta = \frac{\nabla_\theta}{||\nabla_\theta||_1} \tag{6}$$

Next, we apply a standard gradient ascent step on the reward weight vector, $\theta$, using a learning rate $\alpha$: $\theta \leftarrow \theta + \alpha \nabla'_\theta$. After each gradient update, we normalize $\theta$ (Equation 7) to ensure it remains on a consistent scale for the stable training for the policy, $\pi_\psi$, as the semantics of a reward function do not change by scaling (Fu et al., 2018).

$$\theta \leftarrow \frac{\theta}{||\theta||_1} \tag{7}$$

This normalization helps control the magnitude of the weight updates and ensures that the reward and policy optimization remains stable. By iterating the reward updates, reward normalization, and policy learning steps, ELEMENTAL optimizes both the reward function and the policy over a fixed number of iterations.

### 4.3 PHASE 3: REFLECTING

The third phase introduces a self-reflection mechanism, designed to close the loop and iteratively refine the feature function drafted in Phase 1. After Phase 2, the learned policy, $\pi_\psi(s)$, is executed in the environment, and its behavior is compared to the demonstration, $\mathcal{D}$, based on the drafted feature

function $\phi(s)$. Specifically, we calculate the expected feature counts of the generated trajectories under the current policy and the feature counts of the demonstration trajectories (Equation 8).

$$\vec{\Phi}_{\pi_\psi} = \mathbb{E}_{\tau \sim \pi_\psi} \left[ \sum_{s \in \tau} \phi(s) \right], \quad \vec{\Phi}_{\mathcal{D}} = \mathbb{E}_{\tau \sim \mathcal{D}} \left[ \sum_{s \in \tau} \phi(s) \right] \tag{8}$$

Discrepancies between the two feature counts indicate that the current feature function may not fully capture the relevant aspects of the task as demonstrated.

The two feature count vectors are then fed back to the VLM, which uses the feature count differences to revise the feature function $\phi(s)$. By accounting for previously overlooked or misinterpreted features, the VLM's understanding of the task becomes progressively more aligned with the demonstrated behavior. This process of self-reflection continues iteratively, alternating between Phase 2 (reward function and policy optimization) and Phase 3 (feature refinement), allowing the robot to improve its behavior over time.

The reflecting phase is fully automatic, leveraging the policy, $\pi_\psi(s)$, and the feature function, $\phi(s)$, generated in the previous phases. Because both the policy and the feature function are available, ELEMENTAL can continuously refine its understanding of the task without requiring additional input from a human. However, should the user wish to intervene and provide further feedback or corrections, ELEMENTAL could accommodate interaction through prompts in the future work.

## 5 RESULTS

In this section, we evaluate the performance of ELEMENTAL on challenging robotic tasks from IsaacGym (Makoviychuk et al., 2021). We design experiments to demonstrate the effectiveness of ELEMENTAL in both benchmarking against SOTA baselines (Section 5.1) and generalization to task variants (Section 5.2).

**Environments and Tasks**  In benchmark experiments, we test ELEMENTAL and baseline algorithms on nine challenging IsaacGym Robotics tasks, using GPU-accelerated training to enable efficient experiments. These tasks span various domains, including locomotion and manipulation, and are recognized for their complexity in the robot learning community (Makoviychuk et al., 2021). For all methods that utilize a LLM/VLM, we use the OpenAI GPT-4o model unless otherwise noted. We use five demonstrations for each task collected with RL-trained policies.

To the best of our knowledge, this is the first successful application of IRL to IsaacGym – with or without large models – due to the realistic high-dimensional state and action spaces. The performance of ELEMENTAL in these tasks demonstrates its robustness and scalability, making it a suitable framework for solving complex real-world robotic problems using IRL.

**Baselines**  We compare ELEMENTAL against various baselines including LfD methods and LLM-powered reward engineering approaches:

1. LfD methods: We include standard LfD techniques such as Behavior Cloning (BC) and IRL. These baselines learn from demonstration but do not incorporate the VLM feature-extraction or visual input. This comparison demonstrates how traditional LfD methods perform on challenging tasks without access to the feature inference capabilities of VLM.

2. EUREKA: This is the previous SOTA method for reward design with RL in IsaacGym. EUREKA relies on LLMs to infer task features from textual inputs but does not utilize demonstrations, visual inputs, or inverse reinforcement learning.

3. Random Policy: The performance of a random policy establish a lower bound for task performance in each task. This helps quantify the learning improvement made by the other approaches.

4. Ground-Truth (GT) Reward: The performance of a policy trained with GT reward predefined in IsaacGym provides a upper bound of the task performance. Note that although we call it "upper bound", it is not necessarily the maximum performance one can achieve, as the RL is given the same budget of environment steps to train, and the GT reward may not be the best.

5. Ablation of ELEMENTAL: To further analyze the impact of the individual components of ELEMENTAL, we conduct ablation studies on two variants: 1) ELEMENTAL without Self-Reflection: This ablation removes the self-reflection mechanism introduced in Phase 3, keeping

Table 1: This table shows benchmarking results on nine IsaacGym tasks. Bold denotes the best performance except the GT Reward Upper Bound condition.

| Method | IsaacGym Environments | | | | | | | | |
|---|---|---|---|---|---|---|---|---|---|
| | Cartpole | BallBalance | Quadcopter | FrankaCabinet | Ant | Humanoid | Anymal | AllegroHand | ShadowHand |
| Random (LB) | 25.42 | 87.39 | -1.63 | 0.00 | 0.00 | -0.04 | -2.45 | 0.00 | 0.02 |
| BC | 149.85 | 344.55 | -1.19 | 0.01 | -0.05 | -0.43 | -2.14 | 0.04 | 0.03 |
| IRL | 28.15 | 162.06 | -1.87 | 0.00 | 0.88 | 2.13 | -2.22 | 0.01 | 0.01 |
| EUREKA | 215.91 | 454.18 | **-0.22** | 0.21 | 6.88 | 3.78 | -4.24 | 11.12 | 0.00[1] |
| ELEMENTAL (Ours) | 233.92 | **464.40** | -0.30 | **0.36** | **8.49** | **4.70** | **-0.83** | **22.97** | **2.71** |
| w/o Self-Reflection | 114.66 | 153.52 | -0.93 | 0.00 | 5.05 | 3.65 | -1.71 | 0.02 | 0.03 |
| w/o Norm. 1 (Eq. 6) | 186.52 | 423.78 | -1.20 | 0.02 | 7.29 | 2.73 | -0.95 | 13.39 | 2.32 |
| w/o Norm. 2 (Eq. 7) | 192.51 | 459.33 | -0.54 | 0.02 | 3.23 | 4.87 | -1.15 | 0.04 | 1.57 |
| w/o Visual Input | 178.68 | 304.58 | -1.01 | 0.00 | 8.16 | 4.49 | -1.41 | 18.52 | 0.03 |
| w/ Text Demo[2] | 207.51 | 412.17 | -0.92 | 0.00 | 7.43 | 4.60 | -0.88 | 7.07 | 0.04 |
| w/ Rand. Vis. Demo | 269.46 | 352.53 | -1.07 | 0.02 | 7.13 | 3.93 | -1.07 | 20.75 | 2.33 |
| GT Reward (UB) | 260.14 | 461.90 | -0.27 | 0.40 | 7.00 | 5.07 | -0.03 | 23.70 | 0.15 |

Table 2: This table shows reward correlation comparisons on nine IsaacGym tasks.

| Method | IsaacGym Environments | | | | | | | | |
|---|---|---|---|---|---|---|---|---|---|
| | Cartpole | BallBalance | Quadcopter | FrankaCabinet | Ant | Humanoid | Anymal | AllegroHand | ShadowHand |
| EUREKA | 0.77 | -0.53 | 0.96 | 0.93 | 1.00 | 0.59 | -0.86 | 0.34 | 0.00 |
| ELEMENTAL (Ours) | 0.99 | 0.85 | 0.89 | 0.98 | 1.00 | 0.98 | 0.97 | 0.58 | 0.31 |

only the VLM-guided feature inference and policy learning phases; 2) ELEMENTAL without Visual Input: This ablation removes the visual input from the VLM, leaving only text-based language input for feature inference. The comparison with this ablation highlights the importance of visual demonstration in aligning robot behavior with the demonstration.

Performance is evaluated using an average task success rate of the final-100 steps during training, a more reliable metric to assess the overall success compared with the max success during training reported by Eureka. In all experiments, we test with three random seeds and report the best performance, considering the randomness in responses from GPT-4o. Full set of hyperparameters and prompts for ELEMENTAL is provided in supplementary.

## 5.1 BENCHMARKING RESULTS

In the first set of experiments, we benchmark ELEMENTAL and the baselines on the IsaacGym tasks, with the results presented in Table 1. BC performs adequately on simpler tasks such as Cartpole and BallBalance, but fails on more complex tasks due to covariate shift. IRL without VLM-based feature extraction struggles to learn effectively in most tasks, highlighting the challenges posed by IsaacGym's high-dimensional state spaces. While EUREKA is able to learn capable policies, ELEMENTAL achieves on-average $42.3\%$ higher performance and outperforms EUREKA on eight out of nine tasks, demonstrating the effectiveness of integrating IRL with VLM-derived features and visual demonstration information. The learned reward correlation with ground-truth reward shown in Figure 2 also illustrates ELEMENTAL's strong ability to learn well-aligned reward function by matching with demonstrations.

To further evaluate whether VLM/LLMs are more suitable for feature extraction or full reward function drafting, we examine the code execution rates of ELEMENTAL and EUREKA across three algorithm iterations. A higher code execution rate indicates fewer coding errors, suggesting better suitability for language models. As shown in Figure 3, ELEMENTAL achieves a successful code execution rate of approximately 80% in the first iteration, compared to EUREKA's rate of less than 50%. Although both algorithms improve with successive iterations, ELEMENTAL consistently

---

[1]We tried running with six seeds, but Eureka with GPT-4o failed to generate any executable reward function for ShadowHand. In the calculation of percentage improvements over Eureka, we treat the improvements on ShadowHand to be 100%.

[2]As the original implementation in Peng et al. (2024b) is not available, we implement the demonstration in text form with ELEMENTAL.

Table 3: This table compares generalization performance of ELEMENTAL and EUREKA on Ant-variant environments. Results are maximized over three seeds. Bold denotes the best performance.

| Method | Ant Original | w/ Reversed Obs | w/ Lighter Gravity | Ant Running Backward |
|---|---|---|---|---|
| EUREKA | 6.88 | 5.96 | 4.39 | 5.62 |
| ELEMENTAL | **8.49** | **8.47** | **5.89** | **9.30** |
| w/o Visual Input | 8.16 | 7.63 | 3.14 | 7.46 |

generates more executable code. These results suggest that GPT-4o is more effective at feature extraction than at drafting complete reward functions, supporting ELEMENTAL's design choice to offload reward weighting to IRL.

In the ablation studies, ELEMENTAL without Self-Reflection demonstrates reduced performance, highlighting the importance of self-reflection in refining both the feature function through VLM and the policy through IRL. Another interesting comparison is between ELEMENTAL without Self-Reflection and IRL, where both algorithms run a single iteration. The former still outperforms IRL, suggesting that even the initial feature function provides significant benefits. In another ablation, ELEMENTAL without visual inputs, we also observe a decrease in performance compared to full ELEMENTAL, particularly on tasks that are difficult to describe using natural language alone, such as FrankaCabinet and Shadow-Hand.

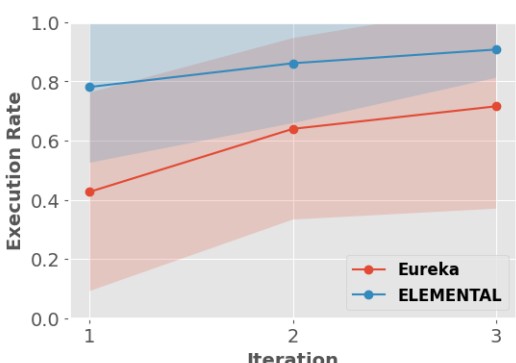

Figure 3: This figure shows a comparison of the code execution rate between ELEMENTAL and EUREKA in the three iterations of the algorithms.

In terms of wall-clock running time, we compared ELEMENTAL and EUREKA under the same computational resource setup. On average across the nine tasks, EUREKA required 68.21 minutes, while ELEMENTAL took 168.36 minutes. The additional time consumed by ELEMENTAL is primarily due to the computational overhead introduced by the IRL updates, which involve environment rollouts to estimate the reward gradient.

## 5.2 Generalization Results

In the second set of experiments, we ask the question "is LLM/VLM just remembering the reward function, as the knowledge cut-off date is Oct 2023, after IsaacGym is public?" To answer this question, we test the generalization capability of ELEMENTAL by applying it to modified versions of the IsaacGym environments, particularly variants of the Ant task. For these modified environments, we change certain properties, such as state vector order, physics property (e.g., gravity coefficient), and the task, to evaluate whether ELEMENTAL's VLM-driven feature inference combined with IRL can adapt to new, unseen environments better than EUREKA. We test on four Ant variants:

1. Ant Original: The standard Ant task without modifications, serving as the baseline environment.

2. Ant with Reversed Observations: The order of the state vector is reversed, testing the algorithms' ability to adapt to changes in the structure of the input data.

3. Ant with Lighter Gravity: The gravity coefficient is reduced from $9.81$ to $5.00$ and requiring the feature and the policy to adjust for a different dynamics. A performance drop is expected as the ant moves with less friction.

4. Ant Running Backward: The task is modified to require the Ant to move backward rather than forward, assessing how well the approaches generalize to a different task objective in the same environment.

We show the comparison between ELEMENTAL and EUREKA in Table 3. In out-of-distribution tasks that language models have not seen in the training set, EUREKA's performance declines, suggesting GPT-4o might have memorized the IsaacGym task rewards, which is not helpful when the state vector is reversed, when the environment dynamics change, or when the task objective is altered. In contrast, ELEMENTAL queries the VLM only for feature functions—information not available in the IsaacGym public data—and uses the demonstration-matching IRL process to determine the reward weights. ELEMENTAL accomplishes an average performance improvement of 41.3%, highlighting its robustness in adapting to changes in both the environment's physical properties and the task's nature.

## 6 CONCLUSION AND FUTURE WORK

We introduced ELEMENTAL, a novel framework that integrates VLMs with LfD, supplements task descriptions with visual demonstrations, and introduces an iterative self-reflection mechanism for robust learning from demonstration. Evaluations showed that ELEMENTAL outperforms previous state-of-the-art methods by 42.3% on standard benchmarks and 41.3% on generalization capabilities in adapting to novel tasks.

While ELEMENTAL demonstrates strong improvements in LfD with VLMs, several avenues for future work remain. First, testing ELEMENTAL in real-world human-robot systems is a critical next step. Real-world scenarios may require addressing the heterogeneity and suboptimality of human demonstrations, and the training wall-clock time may become a bottleneck in human-robot interactions. To address this, more efficient IRL approaches could be explored. Additionally, given the demonstrated utility of visual inputs, future research could investigate alternative methods for providing visual demonstrations, beyond superimposition and key frames.

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
