# ELEMENTAL: INTERACTIVE LEARNING FROM DEMONSTRATIONS AND VISION-LANGUAGE MODELS FOR REWARD DESIGN IN ROBOTICS SUPPLEMENTARY

## 1 PROMPTS

---

**Initial System Prompt.**

```
You are a feature engineer trying to write relevant features
↪   for the reward function to solve
↪   learning-from-demonstration (inverse reinforcement
↪   learning) tasks as effective as possible.

Your goal is to write a feature function for the environment
↪   that will help the agent construct a linear reward
↪   function with the constructed features via inverse
↪   reinforcement learning to accomplish the task described
↪   in text and the demonstration.

Your feature function should use useful variables from the
↪   environment as inputs. The feature function signature
↪   must follow:
@torch.jit.script
def compute_feature(obs_buf: torch.Tensor) -> Dict[str,
↪   torch.Tensor]:
    ...
    return {}

Since the feature function will be decorated with
↪   @torch.jit.script, please make sure that the code is
↪   compatible with TorchScript (e.g., use torch tensor
↪   instead of numpy array).

You should not wrap the function within a class.

Make sure any new tensor or variable you introduce is on the
↪   same device as the input tensors.
```

---

**Initial User Prompt.**

```
The Python environment is
{task_obs_code_string}

Write a feature function for the following task:
↪   {task_description}.

Three keyframes of a demonstration for how to accomplish the
↪   task are shown in the image (superimposing agent pos).
```

---

---

Code output instruction.

The input of the feature function is a torch.Tensor named
↪ `obs_buf` that is a batched state (shape: [batch,
↪ num_obs]).

The output of the feature function should be a dictionary
↪ where the keys are the names of the features and the
↪ values are the corresponding feature values for the
↪ input state.

You must respect the function signature.

The code output should be formatted as a python code string:
↪ "```python ... ```".

Some helpful tips for writing the feature function code:

    (1) You may find it helpful to normalize the features to
    ↪ a fixed range by applying transformations

    (2) The feature code's input variables must be obs_buf:
    ↪ torch.Tensor, which corresponds to the state
    ↪ observation (self.obs_buf) returned by the
    ↪ environment compute_observations() function. Under
    ↪ no circumstance can you introduce new input
    ↪ variables.

    (3) Each output feature should only one a single
    ↪ dimension (shape: [batch]).

    (4) You should think step-by-step: first, think what is
    ↪ important in the task based on the task description
    ↪ and the demonstration and come up with names of the
    ↪ features, then, write code to calculate each feature

    (5) You should be aware that the downstream inverse
    ↪ reinforcement learning only creates reward functions
    ↪ that are linear function of the constructed
    ↪ features; thus, it is important to construct
    ↪ expressive features that humans do care in this task

    (6) Do not use unicode anywhere such as \u03c0 (pi)

---

---

**Self-Reflection prompt.**

```
We trained reward and policy via inverse reinforcement
↪   learning using the provided feature function code with
↪   the demonstration.

We tracked the feature values as well as episode lengths.

The mean values of the last {eval_avg_horizon} steps from
↪   the learned policy are:
{insert}

Please carefully analyze the feedback and provide a new,
↪   improved feature function that can better solve the
↪   task. Some helpful tips for analyzing the feedback:

    (1) If the episode lengths are low, it likely means the
    ↪   policy is unsuccessful

    (2) If the feature counts are significantly different
    ↪   between demo and learned behavior, then this means
    ↪   IRL cannot match this feature with the demo as it is
    ↪   written. You may consider

        (a) Change its scale

        (b) Re-writing the feature: check error in the
        ↪   feature computation (e.g., indexing the
        ↪   observation vector) and be careful about outlier
        ↪   values that may occur in the computation

        (c) Discarding the feature

    (3) If a feature has near-zero weight, the feature may
    ↪   be unimportant. You can consider discarding the
    ↪   feature or rewriting it.

    (4) You may add/remove features as you see appropriate.

Please analyze each existing features in the suggested
↪   manner above first, and then write the feature function
↪   code.
```

## 2 ELEMENTAL HYERPARAMETERS

We tune hyperparameters via a grid search. We summarize the ELEMENTAL hyperparameters in Table 1. All other hyperparameters follow EUREKA's default setup.

| Hyperparameters | Value |
|---|---|
| Reward learning rate $\alpha$ | 1.0 |
| Approximate MaxEnt-IRL number of iterations $m$ | 5 |
| Policy training steps $k$ | 500 |
| Number of algorithm iterations | 3 |
| Code samples generated per iteration | 1 for ShadowHand, 3 for all other tasks |
| Policy Neural Network Architecture | Fully-connected [32, 32] with ReLU activation |

Table 1: Hyperparameters and their values

Table 2: This table compares generalization performance of ELEMENTAL and EUREKA on Ant-variant environments. Results (mean±std) are averaged over three seeds. Bold denotes the best performance.

| Method | Ant Original | w/ Reversed Obs | w/ Lighter Gravity | Ant Running Backward |
|---|---|---|---|---|
| EUREKA | 4.44±2.45 | 4.11 ± 2.07 | 2.94±2.53 | 3.51±3.03 |
| ELEMENTAL | 6.80±1.17 | 7.40±0.98 | 4.35±1.20 | 7.41±1.25 |
| w/o Visual Input | 7.03 ± 1.67 | 7.07 ± 1.07 | 2.57 ± 0.68 | 6.02 ± 2.03 |

# 3 DETAILED RESTULS

## 3.1 GENERALIZATION EXPERIMENT

The mean and standard deviation for generalization experiment across three seeds are shown in Table 2.

## 3.2 CASE STUDY

In this subsection, we present a case study illustrating the iterative process of ELEMENTAL on the Humanoid task. The initial feature function drafted by the Vision-Language Model (VLM) is shown in Box 1. The proposed features—forward_velocity, uprightness, and heading_alignment—are well-aligned with the task objectives of running efficiently while maintaining balance and direction. These features provide a strong starting point for the learning process.

Using this initial feature function, ELEMENTAL trains the IRL process, calculates the feature counts for both the generated trajectories and the demonstration, and feeds this feedback back to the VLM, as shown in Box 2. The feedback reveals key discrepancies, such as lower forward_velocity. Based on this analysis, the VLM revises the feature function, as shown in Box 3. Notably, the revised function introduces a new feature, lateral_velocity, which captures stride consistency by taking the absolute value of the lateral movement. This demonstrates the VLM's capability to construct nonlinear features, expanding the expressiveness of the feature function.

Finally, ELEMENTAL trains the IRL process again using the updated feature function and compiles the feedback. The resulting reward weights assign the highest importance to forward_velocity, with relatively minor contributions from stability-related objectives such as uprightness and lateral_velocity. This distribution aligns well with human intuition for the task, where speed is the primary objective, and stability features serve as secondary constraints.

This case study highlights the potential of ELEMENTAL to produce models at every stage of its pipeline. The feature functions generated by the VLM are human-readable and meaningful, allowing practitioners to inspect and refine them as needed. Additionally, the linear weights learned by ELEMENTAL during the IRL process indicate the relative importance of each feature in matching the demonstration. The iterative self-reflection mechanism ensures that the learned policies and reward functions evolve in alignment with the demonstration.

216
217
218
219
220
221
222
223
224
225
226
227
228
229
230
231
232
233
234
235
236
237
238
239
240
241
242
243
244
245
246
247
248
249
250
251
252
253
254
255
256
257
258
259
260
261
262
263
264
265
266
267
268
269

---

### BOX 1: 1ST ROUND VLM RESULT

To create a feature function aimed at helping a humanoid robot run as fast as possible, we should
↪   focus on features that are most relevant to running speed and ensuring the humanoid's balance
↪   and stability. Useful features could include the forward velocity, balance (uprightness), and
↪   perhaps a component that ensures the humanoid maintains forward momentum without falling.

Here's the feature function based on the observation buffer structure:

```python
import torch
from typing import Dict

@torch.jit.script
def compute_feature(obs_buf: torch.Tensor) -> Dict[str, torch.Tensor]:
    # Extract relevant features from the observation buffer
    # Assuming obs_buf shape: [batch_size, num_obs]

    # Feature: forward velocity (x direction in local frame)
    forward_velocity = obs_buf[:, 1]

    # Feature: uprightness (use the up projection)
    uprightness = obs_buf[:, 10]

    # Feature: heading alignment (use the heading projection)
    heading_alignment = obs_buf[:, 11]

    # Normalize the features
    # Assuming value normalizations such as the velocity range, uprightness [-1, 1], heading
    ↪   alignment [-1, 1]
    max_forward_velocity = 10.0  # This is an arbitrary normalization
    forward_velocity = forward_velocity / max_forward_velocity  # Normalizing forward velocity

    # Combine features into output dictionary
    features = {
        "forward_velocity": forward_velocity,
        "uprightness": uprightness,
        "heading_alignment": heading_alignment,
    }

    return features
```

### Explanation:

1. **`forward_velocity`**: The observation buffer's second column (`obs_buf[:, 1]`) captures the
↪   forward velocity in the local frame.

2. **`uprightness`**: The observation buffer's 11th column (`obs_buf[:, 10]`) captures the upright
↪   projection, which is crucial to ensure the humanoid isn't falling over.

3. **`heading_alignment`**: The observation buffer's 12th column (`obs_buf[:, 11]`) captures heading
↪   alignment, which helps ensure the humanoid is moving forward in the correct direction.

These features will enable the inverse reinforcement learning process to construct a reward function
↪   that emphasizes running fast while maintaining balance and a straight heading.

---

270
271
272
273
274
275
276
277
278
279
280
281
282
283
284
285
286
287
288
289
290
291
292
293
294
295
296
297
298
299
300
301
302
303
304
305
306
307
308
309
310
311
312
313
314
315
316
317
318
319
320
321
322
323

---

**BOX 2: 1ST ROUND FEEDBACK**

```
Calculating mean features on the demonstration results in the following:
episode_lengths: 1000.0
forward_velocity: 0.57
uprightness: 0.94
heading_alignment: 0.91
We trained reward and policy via inverse reinforcement learning using the provided feature function
↪  code with the demonstration.
We tracked the feature values as well as episode lengths.
The mean values of the last 100 steps from the learned policy are:
forward_velocity: 0.4
uprightness: 0.99
heading_alignment: 0.99
IRL reward: 0.05
episode_lengths: 691.47
IRL feature weights: {'forward_velocity': 0.693, 'uprightness': -0.107, 'heading_alignment': -0.201}
Please carefully analyze the feedback and provide a new, improved feature function that can better
↪  solve the task. Some helpful tips for analyzing the feedback:
    (1) If the episode lengths are low, it likely means the policy is unsuccessful
    (2) If the feature counts are significantly different between demo and learned behavior, then
    ↪  this means IRL cannot match this feature with the demo as it is written. You may consider
        (a) Change its scale
        (b) Re-writing the feature: check error in the feature computation (e.g., indexing the
        ↪  observation vector) and be careful about outlier values that may occur in the
        ↪  computation
        (c) Discarding the feature
    (3) If a feature has near-zero weight, the feature may be unimportant. You can consider
    ↪  discarding the feature or rewriting it.
    (4) You may add/remove features as you see appropriate.
Please analyze each existing features in the suggested manner above first, and then write the
↪  feature function code.The input of the feature function is a torch.Tensor named `obs_buf` that
↪  is a batched state (shape: [batch, num_obs]).
The output of the feature function should be a dictionary where the keys are the names of the
↪  features and the values are the corresponding feature values for the input state.
You must respect the function signature.
The code output should be formatted as a python code string: "```python ... ```".

Some helpful tips for writing the feature function code:
    (1) You may find it helpful to normalize the features to a fixed range by applying
    ↪  transformations
    (2) The feature code's input variables must be obs_buf: torch.Tensor, which corresponds to the
    ↪  state observation (self.obs_buf) returned by the environment compute_observations()
    ↪  function. Under no circumstance can you introduce new input variables.
    (3) Each output feature should only one a single dimension (shape: [batch]).
    (4) You should think step-by-step: first, think what is important in the task based on the task
    ↪  description and the demonstration and come up with names of the features, then, write code
    ↪  to calculate each feature
    (5) You should be aware that the downstream inverse reinforcement learning only creates reward
    ↪  functions that are linear function of the constructed features; thus, it is important to
    ↪  construct expressive features that humans do care in this task
    (6) Do not use unicode anywhere such as \u03c0 (pi)
```

---

### BOX 3: 2ND ROUND VLM RESULT

Based on the provided feedback and analysis of how the current features are performing, we can
↪  observe a few key points:

1. **Forward Velocity:**
   - The policy is achieving a lower forward velocity than in the demonstration. This suggests that
   ↪  the current method of calculating or normalizing this feature might need adjustment.
   - The forward velocity weight is quite significant (`0.693`).

2. **Uprightness:**
   - The policy has a higher uprightness than the demonstration. The weight for uprightness is
   ↪  negative (`-0.107`), which may indicate that the model learns to be overly cautious about
   ↪  staying upright, potentially at the expense of moving fast. This negative weight suggests we
   ↪  might need to rethink how this feature is influencing the reward.

3. **Heading Alignment:**
   - The policy has a higher heading alignment than the demonstration. Similar to uprightness, the
   ↪  weight is also negative and relatively significant (`-0.201`), indicating it might be
   ↪  overemphasizing this feature.

Given these points, we aim to improve the feature function by:
1. Adjusting the scaling or normalization of the forward velocity.
2. Revisiting and potentially revising the uprightness and heading alignment features to better
↪  match the demonstration characteristics.
3. Considering new or additional features that might encourage the desired behavior such as
↪  stability or stride consistency.

Here is the improved feature function:

```python
import torch
from typing import Dict

@torch.jit.script
def compute_feature(obs_buf: torch.Tensor) -> Dict[str, torch.Tensor]:
    # Extract relevant features from the observation buffer
    # Assuming obs_buf shape: [batch_size, num_obs]

    # Feature: forward velocity (x direction in local frame)
    forward_velocity = obs_buf[:, 1]

    # Adjust scaling of forward velocity to emphasize higher speeds, assuming range [-1, 1]
    max_forward_velocity = 1.0  # New scaling factor to more accurately reflect achievable max
    ↪  speeds
    forward_velocity = forward_velocity / max_forward_velocity  # Normalizing forward velocity

    # Feature: uprightness (use the up projection)
    uprightness = obs_buf[:, 10]

    # Re-normalize uprightness to give more incentive to be upright but not be overly conservative
    uprightness = (uprightness - 0.5) * 2  # Adjusting the range to new scaling [-1, 1]

    # Feature: heading alignment (use the heading projection)
    heading_alignment = obs_buf[:, 11]

    # Normalize heading alignment to confirm alignment direction
    heading_alignment = (heading_alignment - 0.5) * 2  # Adjusting the range similarly

    # Introducing a new feature: lateral velocity (to ensure stability)
    lateral_velocity = obs_buf[:, 2]
    lateral_velocity = torch.abs(lateral_velocity)  # Use absolute value to avoid
    ↪  penalizing/stabilize sideway motions

    # Normalize lateral velocity, supposing it ranges [-1, 1]
    max_lateral_velocity = 1.0
    lateral_velocity = lateral_velocity / max_lateral_velocity

    # Combine features into output dictionary
    features = {
        "forward_velocity": forward_velocity,
        "uprightness": uprightness,
        "heading_alignment": heading_alignment,
        "lateral_velocity": lateral_velocity
    }

    return features
```

### Explanation:

1. **`forward_velocity`**: Adjusted scaling to make achievable velocities more impactful.
2. **`uprightness`**: Adjusted to a new normalization to try to balance upright positioning without
↪  making the humanoid overly cautious.
3. **`heading_alignment`**: Similar adjustment for normalization.
4. **`lateral_velocity`**: Added a new feature to penalize sideways movement, ensuring stability and
↪  straightforward motion.

These changes aim to provide a more nuanced reward function, hopefully promoting behaviors closer to
↪  the demonstration while balancing speed and stability.

**BOX 4: 2ND ROUND FEEDBACK**

```
Calculating mean features on the demonstration results in the following:
episode_lengths: 1000.0
forward_velocity: 5.67
uprightness: 0.89
heading_alignment: 0.83
lateral_velocity: 2.31
We trained reward and policy via inverse reinforcement learning using the provided feature function
↪   code with the demonstration.
We tracked the feature values as well as episode lengths.
The mean values of the last 100 steps from the learned policy are:
forward_velocity: 5.04
uprightness: 0.65
heading_alignment: 0.92
lateral_velocity: 2.05
IRL reward: 3.54
episode_lengths: 932.42
IRL feature weights: {'forward_velocity': 0.549, 'uprightness': 0.253, 'heading_alignment': -0.079,
↪   'lateral_velocity': 0.119}
Please carefully analyze the feedback and provide a new, improved feature function that can better
↪   solve the task. Some helpful tips for analyzing the feedback:
    (1) If the episode lengths are low, it likely means the policy is unsuccessful
    (2) If the feature counts are significantly different between demo and learned behavior, then
    ↪   this means IRL cannot match this feature with the demo as it is written. You may consider
        (a) Change its scale
        (b) Re-writing the feature: check error in the feature computation (e.g., indexing the
        ↪   observation vector) and be careful about outlier values that may occur in the
        ↪   computation
        (c) Discarding the feature
    (3) If a feature has near-zero weight, the feature may be unimportant. You can consider
    ↪   discarding the feature or rewriting it.
    (4) You may add/remove features as you see appropriate.
Please analyze each existing features in the suggested manner above first, and then write the
↪   feature function code.The input of the feature function is a torch.Tensor named `obs_buf` that
↪   is a batched state (shape: [batch, num_obs]).
The output of the feature function should be a dictionary where the keys are the names of the
↪   features and the values are the corresponding feature values for the input state.
You must respect the function signature.
The code output should be formatted as a python code string: "```python ... ```".

Some helpful tips for writing the feature function code:
    (1) You may find it helpful to normalize the features to a fixed range by applying
    ↪   transformations
    (2) The feature code's input variables must be obs_buf: torch.Tensor, which corresponds to the
    ↪   state observation (self.obs_buf) returned by the environment compute_observations()
    ↪   function. Under no circumstance can you introduce new input variables.
    (3) Each output feature should only one a single dimension (shape: [batch]).
    (4) You should think step-by-step: first, think what is important in the task based on the task
    ↪   description and the demonstration and come up with names of the features, then, write code
    ↪   to calculate each feature
    (5) You should be aware that the downstream inverse reinforcement learning only creates reward
    ↪   functions that are linear function of the constructed features; thus, it is important to
    ↪   construct expressive features that humans do care in this task
    (6) Do not use unicode anywhere such as \u03c0 (pi)
```