# OpenReview forum: "ELEMENTAL: Interactive Learning from Demonstrations and Vision-Language Models for Interpretable Reward Design in Robotics"
_ICLR.cc/2025/Conference — Submitted to ICLR 2025_

### Official Review · Reviewer_62CL · 2024-11-02

**Soundness:** 2
**Presentation:** 3
**Contribution:** 3
**Rating:** 3
**Confidence:** 4

**Summary:**

This paper framework that can utilize a combination of fast simulation, demonstration, and VLMs. The main idea is to (i) use VLMs to useful state features from environment code and image from demonstration, (ii) run IRL algorithm to learn reward and the policy from state features , and (iii) use the state feature matching counts as a reflection metric to compare how the learned policy is close to demonstration. The main contribution of this paper lies in investigating a way to utilize demonstrations and VLMs for improving IRL, which seems a promising direction to pursue.

**Strengths:**

- Idea is intuitive and direction seems promising
- Improved performance from incorporating VLMs into IRL pipeline

**Weaknesses:**

Overall, I like the idea of this paper but it's missing too many results/analysis/discussion to support why & how the method works and what happens during the training.

The main weakness of this paper is that it's only reporting the numbers in main table and does not provide results that can help readers understand how the proposed idea works and that can support the claims made in the paper. For instance,
- Despite the claim, the paper is missing any result or discussion/analysis on the interpretability of rewards, and how it is helpful
- Despite the proposed framework has multiple iterations of training, it's not clear how the performance changes across the iteration, how the *interpretability* of reward improves.
- Is VLM really understanding what demonstration is? What would happen if the VLMs receive sub-optimal data from random exploration in the same tasks? What would happen if you give demonstrations in a different way?
- How crucial it is to give demonstrations in visual observation? what would happen if you give demonstrations as a sequence of states to LLMs instead of using VLMs?
- It's missing analysis/experiments that investigate the effect of VLM choices on the effectiveness of the framework. It could be also nice to include results that show how the method is sensitive to the choice of VLMs. For instance, how good the VLMs should be good to enable this framework to work? Would open-source models be okay?

Also, experiments are missing some details and baselines:
- Details on experimental setup is not clear. Are all the methods using the same resources for training? It's not clear if all the models are trained until convergence.
- It seems to me that BC performance is a bit weak but it's difficult to understand why as there are not that many experimental details. What would be the performance if we use a powerful BC algorithm such as DiffusionPolicy?

Chi, Cheng, et al. "Diffusion policy: Visuomotor policy learning via action diffusion." The International Journal of Robotics Research (2023): 02783649241273668.

**Questions:**

See Weaknesses

---

> ### Author Response · Authors · 2024-11-23
> **Reply to Reviewer 62CL (part 1)**
>
> We thank the reviewer for their thoughtful feedback and for recognizing the promise of the proposed framework. We appreciate the reviewer’s acknowledgment of the intuitive nature of our approach, the potential of combining VLMs and IRL, and the performance improvements. Below, we address the reviewer’s concerns and provide additional details and clarifications to strengthen the claims and analyses presented in the paper. (edited versions of the paper and the supplementary are uploaded with changes in red)
>
> **Q1**: Missing result or discussion/analysis on the interpretability of rewards and how the interpretability improves over multiple iterations.
>
> **A1**: We acknowledge that we did not conduct user studies to explicitly evaluate the interpretability of the learned rewards. However, the design of ELEMENTAL inherently supports interpretability, akin to the argument of interpretability by visual inspection used in InfoGAIL [1] and linear reward interpretability in [2,3]. The Python-code feature functions and the linear combination of features in the reward function allow practitioners to inspect and understand the features and the learned weights, providing transparency into the ELEMENTAL learning process.
>
> To illustrate this interpretability, we included a case study in Supplementary Section 3.2. This case study demonstrates how ELEMENTAL’s self-reflection mechanism identifies task-relevant features and iteratively refines them based on discrepancies between demonstrations and learned policies. In the Humanoid task, ELEMENTAL discovered a new feature, lateral_velocity, during the refinement process. This highlights its ability to generate meaningful, interpretable features that align with the task’s goals.
>
> We believe that this inherent interpretability makes ELEMENTAL a valuable framework for tasks requiring human oversight and refinement. As part of future work, we aim to conduct user studies to further validate and enhance its interpretability in real-world applications.
>
> **Q2**: Is VLM really understanding what demonstration is? What happens with sub-optimal or alternate types of demonstrations? How crucial are visual demonstrations? What if demonstrations are provided as a sequence of states?
>
> **A2**: Thank you for raising these insightful questions. To evaluate the role and impact of visual demonstrations, as well as to assess VLMs’ understanding of demonstrations, we designed two variants of ELEMENTAL:
> - w/ Text Demo: This variant aligns with [4], where demonstrations are provided as text-based state-vector sequences. Since real-world demonstrations often involve thousands of states (each with tens to hundreds of dimensions), directly feeding all states into the LLM would be infeasible. Therefore, following [4]’s setup, we subsample 10 states evenly from the demonstration. These state vectors are then formatted as part of the text input to the VLM.
> - w/ Random Visual Demo: To test whether the VLM effectively extracts task-relevant information from good demonstrations, we replaced the visual demonstration input with random visual demonstrations (e.g., a falling Cartpole or Humanoid). Importantly, the same high-quality demonstrations are still used during the IRL phase; only the VLM visual input is substituted.
>
> The results of these experiments are added to Table 1 (rows “w/ Text Demo” and “w/ Rand. Vis. Demo”). The findings are as follows:
> - Visual Demonstrations vs. Text Demonstrations: The performance with text demonstrations is consistently inferior to visual demonstrations (except for Cartpole), underscoring the importance of rich, contextual information available in visual inputs that text representations may fail to convey.
> - Good vs. Random Visual Demonstrations: ELEMENTAL with good visual demonstrations consistently outperforms the variant using random visual inputs. This indicates that VLMs indeed extract useful task-specific information from high-quality demonstrations. The substantial performance drop with random visual inputs confirms that ELEMENTAL does not merely overfit to arbitrary visual inputs but relies on information embedded.
>
> **Q3**: Analysis on the effect of VLM choices and sensitivity to different models
>
> **A3**: Thank you for this valuable question. Due to time and resource constraints, we have not yet conducted experiments with alternative VLMs to systematically evaluate ELEMENTAL's sensitivity to the choice of VLMs. However, based on findings reported in EUREKA, where performance improves when moving from GPT-3.5 to GPT-4, we hypothesize that ELEMENTAL would exhibit a similar trend. We acknowledge the importance of this analysis and plan to include experiments with open-source and different-quality VLMs in future studies to provide a more comprehensive evaluation of ELEMENTAL's robustness and adaptability.

---

> ### Author Response · Authors · 2024-11-23
> **Reply to Reviewer 62CL (part 2)**
>
> **Q4**: Clarity on experimental setup and training convergence
>
> **A4**: We ensure consistency in computational resources and VLMs (GPT-4o) across all experiments to maintain a fair comparison. Both ELEMENTAL and EUREKA are trained using the same fixed budget of three iterations. The hyperparameters used for all experiments are documented in Supplementary Table 1, providing reproducibility.
>
> **Q5**: Why is BC performance weak? What would be the performance with a more powerful BC algorithm like DiffusionPolicy?
>
> **A5**: Thank you for your suggestion! BC could indeed benefit from more advanced approaches such as DiffusionPolicy. However, to ensure a fair comparison across all learning methods, we maintain a consistent policy architecture throughout the experiments: a two-hidden-layer neural network with 32 units per layer and ReLU activations. This design choice ensures that differences in performance across methods are attributable to the reward design and learning framework rather than discrepancies in model capacity. While DiffusionPolicy may provide stronger results for BC, incorporating it would require an adjustment of architectural baselines across all methods to maintain fairness. We will explore the more powerful BC algorithms in the ELEMENTAL framework in the future work.
>
>
> [1] Li, Y., Song, J., & Ermon, S. (2017). Infogail: Interpretable imitation learning from visual demonstrations. Advances in neural information processing systems, 30.
>
> [2] Sanneman, L., & Shah, J. (2021). Explaining reward functions to humans for better human-robot collaboration. arXiv preprint arXiv:2110.04192.
>
> [3] Sanneman, L., & Shah, J. (2023, March). Transparent value alignment. In Companion of the 2023 ACM/IEEE International Conference on Human-Robot Interaction (pp. 557-560).
>
> [4] Peng, A., Li, B. Z., Sucholutsky, I., Kumar, N., Shah, J. A., Andreas, J., & Bobu, A. (2024). Adaptive language-guided abstraction from contrastive explanations. arXiv preprint arXiv:2409.08212.

---

> > ### Comment · Reviewer_62CL · 2024-11-25
> >
> > Thank you for the response, in particular I appreciate the results on investigating the effect of how you provide demonstrations to the model. But I maintain my score for now as I still think the paper needs an improvement in supporting the main claim on the interpretability of the learned rewards, by assigning more portion of the draft into that. One more point that's not clear to me is: the response to my review mentioned that the paper ensures consistency in computational resources to maintain a fair comparison, but wall clock time for EUREKA and ELEMENTAL is so different from each other (in the response to Reviewer NGpK) as ELEMENTAL takes much more time to train. I'm not sure how this can be considered as a fair comparison.

---

> > > ### Author Response · Authors · 2024-11-27
> > >
> > > Thank you for your response and for recognizing our results investigating the ways to provide demonstrations to the model. We greatly appreciate your insights and constructive feedback.
> > >
> > > Regarding the interpretability of the learned rewards, we acknowledge your concern and have edited the paper to remove explicit claims about interpretability to avoid overstating our findings. We thought we were in keeping given how prior literature had assumed that linear reward functions with semantic features was ipso facto interpretable, but we agree with your feedback that this is inadequate and that a proper empirical characterization would be necessary to make this claim. We have removed those claims and provided the case study about the learned feature and reward functions in the supplementary material to showcase how ELEMENTAL enables iterative feature discovery and refinement. This provides readers with insights into how the framework might support interpretability in future studies involving human users but does not claim that it is interpretable. We believe this adjustment better aligns with the current scope of our contributions while providing a foundation for further exploration.
> > >
> > > Regarding the running time concern, the comparison in the policy performance of ELEMENTAL vs. EUREKA is fair because we give both approaches an equal amount of environment steps to train the RL policies based upon their inferred reward functions. It is true that ELEMENTAL uses more time to predict the reward function, but the wall clock budget allocated for training the policies given the reward function is constant across ELEMENTAL and EUREKA. As demonstrated in our experiments, ELEMENTAL outperforms EUREKA by 42.3% on task success and achieves 41.3% better generalization in out-of-distribution tasks. These results underscore the framework’s potential for robot LfD.
> > >
> > > Thank you once again for your valuable feedback. Your suggestions have helped us improve the clarity of our work and refine its presentation. We hope these clarifications address your concerns and emphasize the positive contributions ELEMENTAL brings to the field.

---

> > > > ### Author Response · Authors · 2024-12-02
> > > >
> > > > Dear Reviewer 62CL,
> > > >
> > > > Thank you once again for your thoughtful and constructive feedback throughout this process. As we near the conclusion of the discussion period, we wanted to follow up to kindly ask whether our responses and the clarifications we provided have helped address your concerns. If there are any remaining questions or aspects of our response that you feel could be improved, we would be more than happy to address them.
> > > >
> > > > Thank you again for your time and for engaging with us during the discussion phase. We deeply appreciate your efforts in reviewing our submission and hope our responses have positively contributed to your perspective on our work.
> > > >
> > > > Best regards,
> > > >
> > > > Authors of ELEMENTAL

---

> > > > > ### Author Response · Authors · 2024-12-03
> > > > >
> > > > > As the discussion period deadline approaches in less than 12 hours, we wanted to thank you once again for your thoughtful feedback and engagement. We hope our responses have positively addressed your concerns, and we remain available to provide any additional clarification for any remaining questions you may have.
> > > > >
> > > > > Thank you for your time and consideration.

---

### Official Review · Reviewer_UW2e · 2024-11-03

**Soundness:** 3
**Presentation:** 3
**Contribution:** 3
**Rating:** 8
**Confidence:** 3

**Summary:**

Using large language models (LLMs) for specifying reward functions has shown success in reinforcement learning. However, existing methods like EUREKA describe tasks and rewards only through language. To remedy this, ELEMENTAL combines learning from demonstration (LfD) and LLMs for reward engineering to extract features, learn a policy and reward, and iteratively refine features. ELEMENTAL leads to better generalization and performance than existing methods.

**Strengths:**

This paper makes a clear extension from EUREKA, by incorporating visual inputs. In addition, it proposes a novel framework of feature extraction, learning, and reflecting. In particular, I find the self-reflection loop to be compelling

The diagram and writing are clear.

**Weaknesses:**

The authors mention that ELEMENTAL helps “align robot behavior with user intentions better” and that EUREKA allows humans to “interpret and interactively refine the robot’s behavior” and is more “user-aligned” (line 144). There are no experiments or further discussion of this, and it is not explored in this work.

Further discussion on the effect of self-reflection, such as the types of features that are discovered through self-reflection, would be interesting.

**Questions:**

1. What is the training time needed for these tasks?
2. Why is Peng et al. (2024b) not included as a baseline?
3. How does this work compare with reinforcement learning from VLM rewards?
4. A more thorough appendix would be useful. For instance, what dimension are the features? What are example features?
5. What is the environment state space for the tasks (line 248)?
6. In Table 2, what are the differences in the EUREKA and ELEMENTAL implementations? For example, are the VLMs / LLMs different, and might that account for the reduced generalization?

---

> ### Author Response · Authors · 2024-11-23
> **Reply to Reviewer UW2e (part 1)**
>
> We thank the reviewer for their thoughtful feedback and positive assessment of our work. We are delighted that the reviewer found our work to be a compelling extension of EUREKA. We are particularly encouraged by the reviewer’s appreciation of our proposed self-reflection loop and its potential for improving feature extraction and task alignment, as well as the clarity of our diagrams and writing. Below, we address the reviewer’s concerns and provide additional details to strengthen our submission. (edited versions of the paper and the supplementary are uploaded with changes in red)
>
> **Q1**: What is the training time needed for these tasks?
>
> **A1**: We have included a comparison of wall-clock running times between ELEMENTAL and EUREKA in Section 5.1. On average across the nine tasks, EUREKA required 68.21 minutes, while ELEMENTAL took 168.36 minutes. The additional time consumed by ELEMENTAL is primarily due to the computational overhead introduced by the IRL updates, which involve environment rollouts to estimate the reward gradient. However, ELEMENTAL’s framework is flexible and can accommodate more efficient IRL methods, such as Adversarial IRL (AIRL), to reduce this overhead. Despite the additional runtime, ELEMENTAL achieves significantly better performance. Our experimental results show that ELEMENTAL outperforms prior work by 24.4% on task success and achieves 41.3% better generalization in out-of-distribution tasks. This performance gain demonstrates that ELEMENTAL's improvements justify its longer runtime, particularly for applications where user-alignment and generalization are critical.
>
> **Q2**: Why is Peng et al. (2024b) not included as a baseline?
>
> **A2**: Thank you for raising this question. Originally, Peng et al. (2024b) was not included because their approach requires demonstrations to be exclusively provided as text-based state-vector sequences. Due to this constraint, their method is only applicable to simpler tasks with short problem horizons (e.g., fewer than ten steps) and where most features are predefined, with only one or two missing. Moreover, their code is not publicly available.
>
> To address the reviewer’s feedback, we created a comparable implementation by subsampling ten states from demonstrations (a scale consistent with Peng et al.'s domains), representing these feature vectors as text inputs. This comparison is included as the "w/ Text Demo" row in Table 1 of the revised paper. Our results demonstrates that ELEMENTAL achieves better performance on more complex IsaacGym environments, and our results highlight the scalability and generalization limitations of Peng et al.'s text-only approach, particularly in tasks requiring higher-dimensional or more diverse inputs.
>
> **Q3**: How does this work compare with reinforcement learning from VLM rewards?
>
> **A3**: Thank you for this excellent question. While reinforcement learning from VLM-generated rewards offers an alternative for specifying objectives, ELEMENTAL provides key advantages that address the challenges inherent in such approaches (noted in the related work section):
> 1. Reward Quality and Calibration: VLM-generated rewards often lack calibration, producing sparse or uncalibrated dense rewards that may misguide fine-grained policy learning. By leveraging IRL, ELEMENTAL directly infers feature weights from demonstrations, ensuring that the learned rewards align closely with desired behaviors and providing dense, accurate learning signals.
> 2. Interpretability: ELEMENTAL allows users to inspect the feature weights learned during IRL, offering an interpretable reward structure. This transparency enables engineers and users to understand, adjust, and refine the task specifications, which is difficult with opaque VLM-generated reward signals.
> 3. Interactivity and Alignment: ELEMENTAL supports iterative refinement through self-reflection, enabling users to interactively guide the robot’s behavior. In contrast, VLM-guided reward approaches lack mechanisms for incorporating iterative feedback or user-driven refinements, which are essential for gaining human trust and improving alignment with user intentions.
>
> These features make ELEMENTAL a robust and user-friendly framework for reward design, particularly in scenarios requiring nuanced and human-aligned behaviors.

---

> ### Author Response · Authors · 2024-11-23
> **Reply to Reviewer UW2e (part 2)**
>
> **Q4**: A more thorough appendix would be useful. For instance, what dimension are the features? What are example features? Further discussion on the effect of self-reflection, such as the types of features that are discovered through self-reflection, would be interesting.
>
> **A4**: Thank you for this helpful suggestion. We have included a detailed case study in Appendix Section 3.2 that illustrates how ELEMENTAL generates and iteratively refines features through its self-reflection mechanism.
>
> The case study focuses on the Humanoid task. As shown in Box1 of the appendix, the VLM proposes three features—forward_velocity, uprightness, and heading_alignment. These features are well-aligned with the task objectives of running while maintaining balance and direction. Using these features, ELEMENTAL performs IRL training, computes feature counts for both generated trajectories and demonstrations, and provides feedback to the VLM based on discrepancies (Box2). In response, as shown in Box3, the VLM introduces a new nonlinear feature: lateral_velocity. This feature captures stride consistency through an absolute value transformation of lateral movement. The updated feature set now consists of four features (forward_velocity, uprightness, heading_alignment, and lateral_velocity). The final reward weights (Box4) assign the highest importance to forward_velocity, with smaller contributions from stability-related features such as uprightness and lateral_velocity. This weighting aligns with human intuition for the task, where speed is the primary goal, and stability features serve as secondary constraints.
>
> This case study highlights ELEMENTAL’s ability to iteratively enhance feature sets by introducing meaningful nonlinear features, moving from three initial features to four. The iterative self-reflection mechanism ensures that the learned policies and reward functions evolve in alignment with the demonstration.
>
> **Q5**: What is the environment state space for the tasks (line 248)?
>
> **A5**: The environment state space corresponds to the observation function of the MDP, which provides information for the VLM to draft based on the environment observations.  For instance, in the Cartpole task, the state space consists of four components: cart position, cart velocity, pole angular position, and pole angular velocity:
>
> ```Python
> class Cartpole(VecTask):
>     """Rest of the environment definition omitted."""
>     def compute_observations(self, env_ids=None):
>         if env_ids is None:
>             env_ids = np.arange(self.num_envs)
>
>         self.gym.refresh_dof_state_tensor(self.sim)
>
>         self.obs_buf[env_ids, 0] = self.dof_pos[env_ids, 0].squeeze()  # cart position
>         self.obs_buf[env_ids, 1] = self.dof_vel[env_ids, 0].squeeze()  # cart velocity
>         self.obs_buf[env_ids, 2] = self.dof_pos[env_ids, 1].squeeze()  # pole angular position
>         self.obs_buf[env_ids, 3] = self.dof_vel[env_ids, 1].squeeze()  # pole angular velocity
>
>         return self.obs_buf
> ```
>
> **Q6**: Differences in EUREKA and ELEMENTAL implementations in Table 2 (now Table 3). Could differences in VLMs/LLMs account for the reduced generalization?
>
> **A6**: Thank you for raising this point. Throughout all experiments, we ensured consistency in experimental conditions for both EUREKA and ELEMENTAL. This includes using identical hyperparameters and the same VLM (GPT-4o) for both methods.
>
> The reduced generalization observed for EUREKA in Table 3 is not due to differences in VLMs or implementation but rather reflects ELEMENTAL’s ability to better leverage demonstrations and iterative refinement to produce task-relevant features and reward functions. In contrast, EUREKA's reliance solely on language-based reward tuning limits its adaptability in unseen environments.
>
> **Q7**: There are no experiments or further discussion of interpretability.
>
> **A7**: We agree that we did not conduct explicit user studies to evaluate whether the learned features and reward functions are interpretable in practice. However, the potential for interpretability in ELEMENTAL is rooted in its design, similar to arguments made for interpretability in InfoGAIL [1]. By leveraging self-reflection and feature-based rewards, ELEMENTAL produces features and reward functions that are human-readable and align well with task objectives, as illustrated in our case study (Supplementary Section 3.2).
>
> [1] Li, Y., Song, J., & Ermon, S. (2017). Infogail: Interpretable imitation learning from visual demonstrations. Advances in neural information processing systems, 30.

---

> ### Comment · Reviewer_UW2e · 2024-11-26
> **Reply to Authors**
>
> Thank you for the helpful responses. I will consider increasing my score to an accept.

---

> > ### Author Response · Authors · 2024-11-27
> >
> > We sincerely thank you for the response and increasing the score!

---

### Official Review · Reviewer_8FuJ · 2024-11-04

**Soundness:** 3
**Presentation:** 3
**Contribution:** 2
**Rating:** 5
**Confidence:** 4

**Summary:**

This paper presents a novel approach to reward tuning that combines language goals and visual user demonstration inputs with Vision-Language Models (VLMs) to address ambiguities inherent in language-only reward tuning. The method consists of three stages. In the first stage, simulator code, language-based goal descriptions, and images of user demonstrations are provided to a VLM, which then generates code to calculate features relevant to the task. In the second stage, a reward function and policy are learned online using maximum-entropy Inverse Reinforcement Learning (IRL). Finally, in the third stage, rollouts from the policy are used to compute the discrepancy between the feature counts in the user demonstrations and those in the actual rollouts. This discrepancy is fed back into the VLM for iterative refinement, alternating between the second and third stages to optimize performance. Experiments conducted on Isaac Gym tasks demonstrate that the proposed method achieves superior performance compared to state-of-the-art (SOTA) language-only reward tuning and IRL methods that do not leverage VLMs.

**Strengths:**

- The proposed method leverages VLMs to propose features relevant to the task rather than generating the entire reward function. This approach minimizes the risk of overfitting to environments encountered during VLM/LLM training and reduces issues with low code execution rates, as confirmed by the paper’s experimental results.
- The proposed framework takes in both a language goal and visual demonstrations, effectively addressing the ambiguities associated with using only one type of input.

**Weaknesses:**

- The requirement for MDP environment code as an input limits this method to simulated environments. In real-world applications, this would require explicitly specifying all relevant objects and dynamics, which could be impractical or infeasible.
- To what extent does the assumption that the reward is a weighted sum of the feature vectors limit the expressiveness of the reward function? This limitation excludes more complex functional forms, such as exponentials, logarithmic functions, or features in the denominator, potentially limiting the method's ability to capture nuanced task-specific details.
- When using a superimposed image as a visual demonstration in tasks like navigation, there is an inherent ambiguity in capturing the temporal direction of actions. For tasks where superimposed images are unsuitable, the method selects approximately four keyframes from the demonstration, introducing an additional need for keyframe identification.
- The method may struggle with highly complex simulation environments, as it requires the entire MDP environment code as input.

**Questions:**

- How does the allowance of up to 3 attempts impact the method's effectiveness? For tasks where the method fails after 3 attempts, would increasing this threshold lead to successful code executions, or would these tasks likely fail at this stage regardless of additional attempts?
- How does the proposed reward in the experiments compare with the ground truth reward? Would be good to see comparisons and analysis on this
- In Table 1, over how many iterations is the proposed method trained? Does each iteration consistently lead to performance improvements? It would be helpful to see reward curves plotted against iteration count to better understand the effectiveness of the reflection stage.

---

> ### Author Response · Authors · 2024-11-23
> **Reply to Reviewer 8FuJ (part 1)**
>
> We thank Reviewer 8FuJ for their thoughtful feedback and constructive suggestions. We are excited that the reviewer recognizes the novelty of ELEMENTAL and its ability to mitigate overfitting risks and effectively leverage both language goals and visual demonstrations. Below, we address the reviewer’s concerns and provide responses to the questions. (edited versions of the paper and the supplementary are uploaded with changes in red)
>
> **Q1**: assumption about MDP environment code is limiting.
>
> **A1**: Thank you for raising this concern. We would like to clarify that ELEMENTAL does not require full MDP code. Instead, the MDP interface is used to define the observation structure that serves as inputs to the reward and policy, enabling the agent to interact with the environment. This design aligns with the standard framework for representing decision-making tasks [1] and is sufficient for guiding VLMs in feature extraction without requiring exhaustive environment details. Further, this assumption is not unique to ELEMENTAL. ELEMENTAL follows this similar pattern as EUREKA and [4] for the dependence of MDP codes.
>
> In real-world applications, constructing MDPs tailored to specific tasks could be feasible and provide transfer within a class of tasks (e.g., household robotics). For instance, frameworks like RoboSuite [2] and Habitat [3] provide abstractions for robotic manipulation and navigation tasks, making it practical to define observations and other components as needed. Moreover, ELEMENTAL’s reliance on visual demonstrations helps alleviate the burden of complex text description about the task by leveraging generic visual inputs to intuitively represent environmental context.
>
> [1] Puterman, M. L. (2014). Markov decision processes: discrete stochastic dynamic programming. John Wiley & Sons.
>
> [2] Zhu, Y., Wong, J., Mandlekar, A., Martín-Martín, R., Joshi, A., Nasiriany, S., & Zhu, Y. (2020). robosuite: A modular simulation framework and benchmark for robot learning. arXiv preprint arXiv:2009.12293.
>
> [3] Savva, M., Kadian, A., Maksymets, O., Zhao, Y., Wijmans, E., Jain, B., ... & Batra, D. (2019). Habitat: A platform for embodied ai research. In Proceedings of the IEEE/CVF international conference on computer vision (pp. 9339-9347).
>
> [4] Peng, A., Li, B. Z., Sucholutsky, I., Kumar, N., Shah, J. A., Andreas, J., & Bobu, A. (2024). Adaptive language-guided abstraction from contrastive explanations. arXiv preprint arXiv:2409.08212.
>
> **Q2**: Expressiveness of reward functions as a linear combination of feature vectors
>
> **A2**: While ELEMENTAL assumes a linear combination of features in the reward function, this does not preclude the expressiveness of the features. Non-linear transformations can be incorporated directly into the feature crafting process, and we explicitly inform the VLM that the reward function is linear, encouraging it to embed non-linear transformations directly into the features (Supplementary Section 1 prompt “initial system prompt”).
>
> We have added a case study in Supplementary Section 3.2 to demonstrate this capability, where the VLM generates a lateral_velocity feature that utilizes a non-linear absolute value transformation to capture stability. By allowing non-linear processing at the feature level, ELEMENTAL retains the ability to model nuanced task-specific details while maintaining the interpretability and simplicity of a linear reward structure.
>
> **Q3**: Ambiguity in capturing temporal direction with superimposed images
>
> **A3**: Thank you for pointing this out. We address the temporal direction ambiguity in superimposed images by using varying color shades to indicate the temporal progression. As shown in Figure 2(a), darker shades represent actions further along in the sequence, providing a visual cue for the direction of motion. We added this clarification in Section 4.1.
>
> **Q4**: The method may struggle with highly complex simulation environments, as it requires the entire MDP environment code as input.
>
> **A4**: As mentioned in A1, ELEMENTAL does not require the entire MDP code; it only utilizes the observation function to guide the VLM in interpreting and extracting features. Our results demonstrate ELEMENTAL’s effectiveness in highly complex environments, such as AllegroHand (88-dimensional observation space) and ShadowHand (211-dimensional observation space). Despite the complexity of these observation spaces, ELEMENTAL consistently outperforms EUREKA (more than 100% improvements), as shown in Tables 1 and 2. Notably, EUREKA fails to generate any executable reward function for ShadowHand, highlighting ELEMENTAL’s advantage in leveraging VLMs to synthesize meaningful and task-relevant features in complex scenarios.

---

> ### Author Response · Authors · 2024-11-23
> **Reply to Reviewer 8FuJ (part 2)**
>
> **Q5**: How does the allowance of up to 3 attempts impact the method's effectiveness? Would increasing this threshold lead to successful code executions, or would these tasks likely fail at this stage regardless of additional attempts?
>
> **A5**: In our experiments across the nine IsaacGym environments, ELEMENTAL achieved success within three attempts in all cases, which is why we reported results based on this threshold. This setup aligns with EUREKA’s experiment protocol as described in their paper. While we observed no need to exceed three attempts in the tested scenarios, we hypothesize that in significantly more complex environments, increasing the number of iterations could provide additional opportunities for ELEMENTAL to refine its feature functions and would likely not hurt performance.
>
> **Q6**: How does the proposed reward in the experiments compare with the ground truth reward?
>
> **A6**: Thank you for this insightful question. We have added Table 2 in the revised paper, which compares the reward correlations between the learned reward functions and the ground truth reward across nine IsaacGym tasks. ELEMENTAL achieves consistently higher reward correlations than EUREKA, highlighting its strong ability to infer well-aligned reward functions through demonstration matching.
>
> **Q7**: In Table 1, over how many iterations is the proposed method trained? Does each iteration consistently lead to performance improvements?
>
> **A7**: Both ELEMENTAL and EUREKA are trained over three iterations. While plotting reward curves across iterations is challenging due to the varying scales of success metrics across domains, the effectiveness of the reflection stage is evident from both quantitative and qualitative analyses. In BallBalance, rewards improved from 44.03 to 88.65 to 464.40 over the three iterations, demonstrating consistent performance gains. Additionally, as shown in Figure 3, the code execution rate improves with iterations, reflecting the increasing quality of the generated feature functions. To further illustrate the impact of self-reflection, we have provided a detailed case study in Supplementary Section 3.2, showing how the feature functions evolve and improve through iterative feedback.

---

> > ### Comment · Reviewer_8FuJ · 2024-11-27
> >
> > Thank you for the detailed rebuttal. I appreciate the additional clarifications and experiments provided.
> >
> > I will maintain my score as it is, as requiring code access to either the full MDP or just the observation space inherently constrains this work to a lab setting where such access can be assumed. While current robotics benchmarks may provide this real-world abstraction, this assumption does not hold outside of controlled lab or benchmark environments, limiting the broader applicability of the approach.

---

> ### Author Response · Authors · 2024-11-28
>
> Thank you for your thoughtful feedback and for acknowledging the detailed rebuttal and additional clarifications we provided. We greatly appreciate the opportunity to address your concerns and the constructive engagement in this discussion. We hope the reviewer will engage with us in good faith and evaluate the overall contribution and our improvements to the paper and discussion in the rebuttal. We would kindly ask the reviewer to please reconsider their assessment based upon precedent in prior work, including at ICLR.
>
> Our reliance on environment code for the observation space is a common assumption in prior works published at top-tier venues, including EUREKA, published at ICLR [1], and Peng et al. (2024), published at CoRL [2]. This assumption has been widely accepted as a practical building block for advancing methods toward real-world applications. Additionally, as robotics research continues to leverage structured benchmarks and simulations, it is not unreasonable to envision the development of dictionaries or libraries of code for common environments, which would further support the generalization of methods like ours to broader settings.
>
> Finally, we appreciate the time and effort you invested in evaluating our work and providing valuable feedback. Your insights have been instrumental in refining our presentation of ELEMENTAL within the broader context of robotics and learning from demonstration. Thank you once again for your thoughtful critique and for helping us improve the clarity of our work.
>
> [1] Ma, Y. J., Liang, W., Wang, G., Huang, D. A., Bastani, O., Jayaraman, D., ... & Anandkumar, A. Eureka: Human-Level Reward Design via Coding Large Language Models. In The Twelfth International Conference on Learning Representations.
>
> [2] Peng, A., Li, B. Z., Sucholutsky, I., Kumar, N., Shah, J., Andreas, J., & Bobu, A. Adaptive Language-Guided Abstraction from Contrastive Explanations. In 8th Annual Conference on Robot Learning.

---

> > ### Author Response · Authors · 2024-12-02
> >
> > Dear Reviewer 8FuJ,
> >
> > Thank you once again for your thoughtful and constructive feedback throughout this process. As we near the conclusion of the discussion period, we wanted to follow up to kindly ask whether our responses and the clarifications we provided have helped address your concerns. If there are any remaining questions or aspects of our response that you feel could be improved, we would be more than happy to address them.
> >
> > Thank you again for your time and for engaging with us during the discussion phase. We deeply appreciate your efforts in reviewing our submission and hope our responses have positively contributed to your perspective on our work.
> >
> > Best regards,
> >
> > Authors of ELEMENTAL

---

> > > ### Author Response · Authors · 2024-12-03
> > >
> > > As the discussion period deadline approaches in less than 12 hours, we wanted to thank you once again for your thoughtful feedback and engagement. We hope our responses have positively addressed your concerns, and we remain available to provide any additional clarification for any remaining questions you may have.
> > >
> > > Thank you for your time and consideration.

---

### Official Review · Reviewer_NGpk · 2024-11-04

**Soundness:** 3
**Presentation:** 4
**Contribution:** 2
**Rating:** 5
**Confidence:** 4

**Summary:**

The paper proposes "ELEMENTAL", an approach for integrating user demonstrations in language-model-based reward specifications such as in EUREKA. The paper argues that such demonstrations could alleviate the ambiguity of language task specifications. The approach has 3 phases. The first one involves prompting the VLM with text and demonstrations to obtain an executable feature function. In this stage, the demonstration is represented as either a superimposed image or four keyframes. In the second stage, the aim is to learn a reward function that is linear in the features and tries to match the demonstrations, and in an inner loop updates the policy via PPO and using the obtained reward. In the last stage, the agent reflects on its feature function using the discrepancy of the feature counts in the dataset and policy-generated trajectories. The paper includes several simulation-based evaluations of the method and mainly compares it to EUREKA as a baseline. The results show a consistently significant improvement in performance in comparison to EUREKA.

**Strengths:**

- the paper is very well written and self-contained. It's also quite a smooth read.
- the proposed approach is quite interesting and using demonstrations to alleviate language ambiguity is a needed step for automatic reward design.
- the experiments in simulation are quite extensive and the results support the main claims of the paper.
- the experiments include important ablations of some aspects of the method.

**Weaknesses:**

- the paper lacks motivation for the choice of an IRL-based approach (with reward linear in the features) to include visual inputs as opposed to following the EUREKA-style approach.
- the paper is only validated in simulation. It would be interesting to see whether these approaches could alleviate the reward engineering usually required to handle real-world problems such as jerky motions and unsafe behaviors. Just one real-world experiment with a robot would suffice. For instance, looking at your simulation environments, an experiment with either the Franka, ANYmal, or ShawdowHand would be a great addition to the evaluation. This could also be a sim-to-real transfer experiment.
- the paper lacks ablations of the various normalization steps (equations 6 and 7).
- the paper does not include any information on the amount of time needed for a full run of the algorithm on the various tasks. I think this aspect is very important for readers to decide whether the approach is feasible for their applications and for future methods to improve upon this. I would suggest comparing the wallclock time of running your method and EUREKA.

I am willing to raise my score if these points are properly addressed.

**Questions:**

- how is keyframe selection performed for the manipulation tasks? This aspect is important to understand the assumptions you make (do you assume to have access to such keyframes from an oracle/user for each demonstration?) or do you have some method to get them (very important for reproducibility)?
- In Table 2, is this the reward or success rate or what? why is it missing a standard deviation? why is there such a difference between ant original and ant with reversed observation? this difference might indicate that the statistical significance of the claims based on this table is questionable. Please update the paper (text and table caption) to make the metric clearer.
- how important/necessary are the normalization steps?
- ELEMENTAL without visual inputs is consistently worse than EUREKA, which brings the question of whether a different approach to reward design (not IRL-based) is more suitable for this kind of problem. Can you please discuss this?
- intuitively, what would be the advantage of this method versus using a VLM itself as a success detector? success detectors like the ones presented in [1] could in theory also receive textual instructions and demonstrations. I agree that they would give out a sparse reward signal, or if they are modified to produce denser rewards this value would be quite uncalibrated, but I am interested in the authors' opinion on this matter.

[1] Du, Yuqing, et al. "Vision-language models as success detectors." arXiv preprint arXiv:2303.07280 (2023).

---

> ### Author Response · Authors · 2024-11-23
> **Reply to Reviewer NGpK (part 1)**
>
> We thank the reviewer for their thoughtful feedback and for recognizing the paper’s contributions, particularly the well-written presentation, the novel approach of using demonstrations to alleviate language ambiguity, and the extensive experiments supporting the main claims. Below, we address the reviewer’s concerns. (edited versions of the paper and the supplementary are uploaded with changes in red)
>
> **Q1**: Motivation for IRL-based approach (with reward linear in the features)
>
> **A1**: One of the significant challenges in language-based reward design (such as in EUREKA) is determining the relative importance or weights of the features in the reward function, which we noted in the paper’s introduction. This process involves subtle mathematical trade-offs, which language models are not inherently well-equipped to handle. IRL, on the other hand, excels in this aspect by leveraging demonstrations to infer feature weights that match the desired behavior. Additionally, the linear form of the reward function provides possible interpretability, allowing engineers to inspect and refine the learned feature weights, which is beneficial for debugging and task refinement. In contrast, EUREKA-style approaches rely entirely on language models to specify the full reward function, which introduces challenges such as overfitting to training environments, lower code execution rates (illustrated in Figure 3), and reduced generalization to new tasks (demonstrated in Table 3). By integrating IRL with feature-based reward inference, ELEMENTAL addresses these limitations, combining the strengths of IRL for precise weight inference and VLMs for feature extraction.
>
> **Q2**: Ablations of the normalization steps (Equations 6 and 7)
>
> **A2**: Thank you for highlighting this point. We have now included ablation studies on the normalization steps in Table 1. The results demonstrate that removing either normalization step significantly impacts performance across tasks:
> - w/o Norm. 1 (Eq. 6): Removing gradient normalization results in learning instability, particularly in tasks requiring precise policy adjustments, such as Quadcopter and FrankaCabinet.
> - w/o Norm. 2 (Eq. 7): Omitting weight normalization leads to inconsistent reward scaling, which adversely affects policy learning stability. This is evident from diminished performance in environments like AllegroHand and ShadowHand.
> These findings highlight the crucial role of both normalization steps in stabilizing the learning process and ensuring robust performance across diverse tasks.
>
> **Q3**: Wall-clock running time
>
> **A3**: Thank you for raising this important consideration. We have now included a comparison of wall-clock running times between ELEMENTAL and EUREKA in Section 5.1. On average across the nine tasks, EUREKA required 68.21 minutes, while ELEMENTAL took 168.36 minutes. The additional time consumed by ELEMENTAL is primarily attributed to the computational overhead introduced by the IRL updates, which involve environment rollouts to estimate the reward gradient. While ELEMENTAL's IRL component increases runtime, it is important to note that the framework is flexible and can accommodate more efficient IRL methods, such as Adversarial IRL (AIRL), which could help alleviate the computational overhead. This flexibility ensures that ELEMENTAL can be adapted to meet the time constraints of various applications. Despite the additional runtime, ELEMENTAL achieves significantly better performance. Our experimental results show that ELEMENTAL outperforms prior work by 24.4% on task success and achieves 41.3% better generalization in out-of-distribution tasks. This performance gain demonstrates that ELEMENTAL's improvements justify its longer runtime, particularly for applications where user-alignment and generalization are critical.

---

> ### Author Response · Authors · 2024-11-23
> **Reply to Reviewer NGpK (part 2)**
>
> **Q4**: Keyframe selection for manipulation tasks
>
> **A4**: For the manipulation tasks, an oracle provided the keyframes based on the demonstration. For instance, as shown in Figure 2, we provided images of key stages such as reaching the cabinet handle, opening the cabinet, and the final stage with the cabinet opened. These keyframes were selected to ensure that critical task stages were represented visually. As noted in the future work section (Section 6), we acknowledge that this reliance on manually selected keyframes could be a limitation. Future research could explore alternative, such as automated keyframe extraction [1-3].
>
> [1] Piperagkas, G. S., Mariolis, I., Ioannidis, D., & Tzovaras, D. (2017). Key-frame extraction with semantic graphs in assembly processes. IEEE Robotics and Automation Letters, 2(3), 1264-1271.
>
> [2] Chen, L., & Wang, Y. (2017). Automatic key frame extraction in continuous videos from construction monitoring by using color, texture, and gradient features. Automation in Construction, 81, 355-368.
>
> [3] Yue, H., Yu, Y., Wu, X., & Chen, W. (2018, May). Keyframe extraction and loop closure detection considering robot motion. In 2018 13th IEEE Conference on Industrial Electronics and Applications (ICIEA) (pp. 847-851). IEEE.
>
> **Q5**: Clarification of Table 2 (now Table 3)
>
> **A5**: The metric presented in Table 3 is the success rate, consistent with Table 1. As mentioned in Section 5 (right before Section 5.1), "we test with three random seeds and report the best performance, considering the randomness in responses from GPT-4o," and therefore standard deviations were not originally included. Upon revisiting the results, we identified an error in Table 3 where average values were used instead of the best performance value for some entries. We apologize for this error and thank the reviewer for pointing out the issue. This error has been corrected in Table 3, and the main paper has been updated accordingly. The results now show that ELEMENTAL outperforms EUREKA by 24.4% on task success and achieves 41.3% better generalization in out-of-distribution tasks.
>
> We have additionally added Supplementary Table 2, which reports the mean ± standard deviation across three random seeds. These updated results clearly demonstrate that ELEMENTAL achieves better generalization performance compared to EUREKA across all Ant-variant environments. For example, in the challenging "Ant Running Backward" environment, ELEMENTAL achieves a success rate of 7.41 $\pm$ 1.25 compared to EUREKA's 3.51 $\pm$ 3.03.
>
> We hope these updates help clarify the metrics, reinforce the validity of our claims, and emphasize ELEMENTAL’s superior generalization capabilities over EUREKA.
>
> **Q6**: ELEMENTAL w/o visual inputs performs worse than EUREKA. Are none-IRL-based approaches more suitable?
>
> **A6**: Thank you for this insightful question. We believe that EUREKA’s strong performance in standard IsaacGym tasks may stem from the memorization of these environments, as GPT-4o, the backbone for EUREKA, could have encountered these tasks during its training. This memorization likely gives EUREKA an advantage when it comes to standard benchmarks but limits its ability to generalize to out-of-distribution tasks. As mentioned by Reviewer 8FuJ, ELEMENTAL’s approach minimizes the risk of overfitting to environments encountered during VLM/LLM training. To investigate this, we conducted additional experiments evaluating ELEMENTAL w/o visual inputs inputs on the generalization tasks. The results are summarized in Table 3 and Supplementary Table 2. ELEMENTAL without visual inputs consistently performs on par with or outperforms EUREKA across most Ant-variant environments (on average 47.3% higher performance). These results suggest that IRL-based reward design, even without visual inputs, is a competitive paradigm compared to RL-based approaches like EUREKA. By leveraging demonstration matching, ELEMENTAL can infer reward weights that generalize better to unseen environments. While incorporating visual inputs significantly enhances ELEMENTAL’s performance, the IRL framework alone proves robust and effective in adapting to novel scenarios.

---

> ### Author Response · Authors · 2024-11-23
> **Reply to Reviewer NGpK (part 3)**
>
> **Q7**: Advantage of ELEMENTAL over VLM-based success detectors
>
> **A7**: Thank you for this excellent question. We believe ELEMENTAL provides key advantages over VLM-based success detectors such as those presented in [1]. Success detectors directly assess task-state alignment and can provide sparse reward signals or uncalibrated dense rewards. However, as noted in the LM-Assisted Robot Learning paragraph of the related work section, these approaches face several challenges:
> 1. Reward Quality and Calibration: the rewards generated by VLMs often lack calibration and can be uninformative or misleading for fine-grained policy learning. ELEMENTAL, by contrast, leverages IRL to infer feature weights and rewards directly from demonstrations, ensuring that the learned rewards align more closely with desired behavior.
> 2. Interpretability: ELEMENTAL allows for the inspection of feature weights learned during IRL, providing an interpretable reward structure that can be adjusted or refined by engineers or users.
> 3. Interactivity and Alignment: ELEMENTAL supports iterative refinement through self-reflection, enabling users to interactively influence the robot's behavior. In contrast, VLM-guided policy training lacks mechanisms for user-driven refinement and feedback.
>
> **Q8**: Validation in real-world experiments
>
> **A8**: Thank you for this suggestion! We agree that validating ELEMENTAL in real-world scenarios would be an important next step to demonstrate its applicability beyond simulation environments. We are actively working to demonstrate ELEMENTAL’s capability in real robots.

---

> ### Comment · Reviewer_NGpk · 2024-11-24
> **Reply to Authors**
>
> Thank you for the rebuttal. The authors answered my questions with a great deal of detail. However, I will keep my score as it it due to 2 main limitations of the work. The first one is its reliance on keyframing, which is not a trivial assumption to eliminate via automation. The second limitation is the significant increase in wall-clock time compared to previous work, limiting the applicability of the proposed approach to real-world scenarios.

---

> > ### Author Response · Authors · 2024-11-27
> >
> > Thank you for your thoughtful feedback and recognizing the detail and effort in our rebuttal. We seek below to respond to your remaining concerns and politely ask that you might reconsider your position given that we have positively addressed many of your comments and that, despite the noted limitations of our work, that our approach has strong intellectual merit as the first IRL approach to incorporate VLMs for reward function engineering and that we have achieved positive results over prior work in terms of the quality of the learned behaviors.
> >
> > Regarding keyframing, we acknowledge this as a limitation for manipulation tasks, but our locomotion tasks do not require keyframing. Our framework for locomotion tasks uses superimposed images automatically constructed with ten images sampled from demonstration with equal distance, and on these domains ELEMENTAL achieves 23.0% performance gain over EUREKA. Importantly, ELEMENTAL is the first framework to demonstrate the effectiveness of combining VLMs and IRL to leverage demonstration image representations for LfD. This represents a novel step forward in robotic learning and opens the door to further advancements.
> >
> > Regarding wall-clock time, ELEMENTAL achieves superior performance in learning the target behavior by trading wall clock time for more intelligent reasoning about the underlying task specification conveyed through the natural language- and demonstration-based inputs. IRL approaches naturally require additional because they infer both the reward function and the policy from user demonstrations and align robot behavior with user intentions. This ability to infer reward functions from demonstrations is a key strength of IRL and makes it a distinct approach and interface for users to program robots. While ELEMENTAL’s current runtime is longer, we believe its capacity to leverage VLMs and IRL for learning from demonstration sets it apart and highlights its potential. Our experimental results underscore ELEMENTAL’s performance advantages: it outperforms EUREKA by 42.3% on task success and achieves 41.3% better generalization in out-of-distribution tasks. These results demonstrate that ELEMENTAL offers significant improvements in aligning robot behavior with desired outcomes. Future research can address the runtime challenge by adopting more sample-efficient IRL algorithms, such as Adversarial IRL (AIRL), which require fewer environment rollouts for reward gradient estimation.
> >
> > While we recognize the limitations, we respectfully argue that the value of ELEMENTAL lies in its novel integration of VLMs and IRL, demonstrating the feasibility and potential of this approach for robot learning. Thank you once again for your thoughtful critique and for acknowledging the contributions of our work. Your feedback has been invaluable in helping us refine and improve the presentation of ELEMENTAL. Would you please consider revising your score higher based upon the improvements we have made in the paper in response to your feedback as well as our supporting arguments here?

---

> > > ### Author Response · Authors · 2024-12-02
> > >
> > > Dear Reviewer NGpK,
> > >
> > > Thank you once again for your thoughtful and constructive feedback throughout this process. As we near the conclusion of the discussion period, we wanted to follow up to kindly ask whether our responses and the clarifications we provided have helped address your concerns. If there are any remaining questions or aspects of our response that you feel could be improved, we would be more than happy to address them.
> > >
> > > Thank you again for your time and for engaging with us during the discussion phase. We deeply appreciate your efforts in reviewing our submission and hope our responses have positively contributed to your perspective on our work.
> > >
> > > Best regards,
> > >
> > > Authors of ELEMENTAL

---

> > > > ### Author Response · Authors · 2024-12-03
> > > >
> > > > As the discussion period deadline approaches in less than 12 hours, we wanted to thank you once again for your thoughtful feedback and engagement. We hope our responses have positively addressed your concerns, and we remain available to provide any additional clarification for any remaining questions you may have.
> > > >
> > > > Thank you for your time and consideration.

---

### Meta-Review · Area_Chair_HFfh · 2024-12-20

**Metareview:**

The paper presents a framework for integrating vision-language models with inverse reinforcement learning to enhance reward design and generalization in robotic tasks.

Strengths:

- The proposed approach offers an extension to existing reward design methods, addressing ambiguities in language-only reward tuning using demonstrations.
- Promising experimental results show the method's superior generalization and performance compared to the baseline.
- Self-reflection and iterative refinement provide an interesting method for improving reward function alignment.

Weaknesses:

- Dependency on MDP environment code restricts the framework to controlled lab settings, raising concerns about broader applicability.
- Claims of interpretability remain insufficiently substantiated.
- Lack of analysis and experiments to justify the effectiveness of the design choices.
- High computational requirements and reliance on keyframing limit the practicality of the method in real-world scenarios.

While ELEMENTAL demonstrates intellectual merit, the reviewers agree that the novelty is limited and the paper lacks sufficient empirical validation to justify its claims of real-world applicability and interpretability. Addressing these limitations through more comprehensive experiments and theoretical support would significantly enhance its contribution. Given the mixed reviews and critical weaknesses identified, I recommend rejection in the current form.

**Additional Comments On Reviewer Discussion:**

During the rebuttal, reviewers highlighted concerns about MDP dependency, insufficient evidence for interpretability, and practical limitations like computational demands and keyframing reliance. The authors added clarifications, experiments, and supplementary material, addressing some but not all concerns. While the intellectual merit was acknowledged, issues around real-world applicability and interpretability remain unresolved.

---

### Decision · Program_Chairs · 2025-01-22

Reject